# Inhibition of the sodium-dependent HCO₃⁻ transporter SLC4A4, produces a cystic fibrosis-like airway disease phenotype

Vinciane Saint-Criq[1†], Anita Guequén[2,3], Amber R Philp[2,3], Sandra Villanueva[2], Tábata Apablaza[2,3], Ignacio Fernández-Moncada[2], Agustín Mansilla[2,3], Livia Delpiano[1], Iván Ruminot[2,4], Cristian Carrasco[5], Michael A Gray[1], Carlos A Flores[2,4]*

[1]Biosciences Institute, The Medical School, Newcastle University, Newcastle upon Tyne, United Kingdom; [2]Centro de Estudios Científicos, Valdivia, Chile; [3]Universidad Austral de Chile, Valdivia, Chile; [4]Universidad San Sebastián, Valdivia, Chile; [5]Subdepartamento de Anatomía Patológica, Hospital Base de Valdivia, Valdivia, Chile

*For correspondence: cflores@cecs.cl; carlos.flores@uss.cl

Present address: †Université Paris-Saclay, INRAE, AgroParisTech, Micalis Institute, Jouy-en-Josas, France

Competing interest: The authors declare that no competing interests exist.

**Abstract** Bicarbonate secretion is a fundamental process involved in maintaining acid-base homeostasis. Disruption of bicarbonate entry into airway lumen, as has been observed in cystic fibrosis, produces several defects in lung function due to thick mucus accumulation. Bicarbonate is critical for correct mucin deployment and there is increasing interest in understanding its role in airway physiology, particularly in the initiation of lung disease in children affected by cystic fibrosis, in the absence of detectable bacterial infection. The current model of anion secretion in mammalian airways consists of CFTR and TMEM16A as apical anion exit channels, with limited capacity for bicarbonate transport compared to chloride. However, both channels can couple to SLC26A4 anion exchanger to maximise bicarbonate secretion. Nevertheless, current models lack any details about the identity of the basolateral protein(s) responsible for bicarbonate uptake into airway epithelial cells. We report herein that the electrogenic, sodium-dependent, bicarbonate cotransporter, SLC4A4, is expressed in the basolateral membrane of human and mouse airways, and that it's pharmacological inhibition or genetic silencing reduces bicarbonate secretion. In fully differentiated primary human airway cells cultures, SLC4A4 inhibition induced an acidification of the airways surface liquid and markedly reduced the capacity of cells to recover from an acid load. Studies in the *Slc4a4*-null mice revealed a previously unreported lung phenotype, characterized by mucus accumulation and reduced mucociliary clearance. Collectively, our results demonstrate that the reduction of SLC4A4 function induced a CF-like phenotype, even when chloride secretion remained intact, highlighting the important role SLC4A4 plays in bicarbonate secretion and mammalian airway function.

## Editor's evaluation

This paper is of interest to scientists and clinicians within the field of muco-obstructive diseases in the airways, such as cystic fibrosis (CF) and chronic obstructive pulmonary disease (COPD). It identifies the sodium-bicarbonate cotransporter SLC4A4 as a key component of the mechanism by which normal airways prevent the formation of sticky mucus and defend their selves against bacterial and viral infections.

## Introduction

Bicarbonate (HCO$_3^-$) and chloride (Cl$^-$) are actively secreted into the lumen of airways by the lining epithelial cells. Even though, the pathophysiological consequences of decreased secretion of these anions has been extensively documented in cystic fibrosis (CF), the most common, autosomal recessive, disease in humans, there is still much discussion whether HCO$_3^-$ secretion per se affects airway homeostasis (*Zajac et al., 2021*; *Morrison et al., 2022*). Impaired Cl$^-$ secretion reduces the volume of the fluid that covers the airway epithelium, the airway surface liquid (ASL), leading to ciliary dysfunction and promoting mucus stasis and airway obstruction (*Cantin et al., 2006*; *Saint-Criq and Gray, 2017*). Deficient HCO$_3^-$ secretion reduces ASL pH which compromises post-secretory mucin maturation and clearance (*Hoegger et al., 2014*), impairs the antimicrobial function of epithelial cells (*Pezzulo et al., 2012*; *Shah et al., 2016*) and increases fluid absorption, further decreasing ASL hydration (*Garland et al., 2013*). Whilst direct measurement of pH in the distal airways of CF children showed no acidification of the ASL (*Schultz et al., 2017*), it has been demonstrated that the addition of HCO$_3^-$ induced an increase in ASL height in human airway epithelial cells (hAECs) cultures and restored the normal properties of mucus from CF patients (*Garland et al., 2013*; *Stigliani et al., 2016*). Moreover, the use of aerosolized HCO$_3^-$ into CF-pig airways increased bacterial killing, clearly indicating that HCO$_3^-$ supplementation can correct inherent defects of CF in the airways (*Pezzulo et al., 2012*; *Kim et al., 2021*).

Even though, impaired HCO$_3^-$ secretion is recognised as a detrimental component of CF airway disease, a full mechanistic understanding of the process and players involved in transcellular HCO$_3^-$ transport in the airways is lacking. In normal airways, HCO$_3^-$ is secreted by CFTR and TMEM16A channels, but after inflammatory signalling, HCO$_3^-$ secretion is augmented through increased expression of the Cl$^-$/HCO$_3^-$ exchanger Pendrin (SLC26A4) (*Han et al., 2016*; *Poulsen et al., 1994*; *Kim et al., 2019*; *Rehman et al., 2020*; *Garnett et al., 2011*). Importantly, the mechanism of basolateral HCO$_3^-$ transport/uptake in native pulmonary tissues is still lacking. We reasoned that the identification and functional inhibition of such basolateral membrane proteins could be used as a proof-of-concept to better understand the role of HCO$_3^-$ secretion in airway homeostasis without altering Cl$^-$ secretion. Here using in vitro fully differentiated human bronchial epithelial cells, we identified the Na$^+$-dependent, HCO$_3^-$ electrogenic cotransporter (NBCe1), SLC4A4, as the basolateral protein responsible for HCO$_3^-$ influx, which couples to apical CFTR for HCO$_3^-$ secretion into the ASL. Importantly, SLC4A4 inhibition induced acidification of the ASL, revealing the pivotal role of this cotransporter in airway pH homeostasis. These observations were further tested in murine models, and revealed that SLC4A4 is involved in both basal and Ca$^{2+}$-stimulated HCO$_3^-$ secretion in mice airways. Strikingly, an *Slc4a4*$^{-/-}$ mouse model showed significant pathological signs of muco-obstructive disease and reduced mucociliary clearance, confirming that inhibition of HCO$_3^-$ secretion alters airway homeostasis, mimicking what has been observed in human CF, and identifying a critical role played by SLC4A4.

## Results

### Primary human airway epithelial cells express bicarbonate transporters of the SLC4A family

We first investigated, by PCR, whether primary hAECs expressed different members of the SLC4A family of Na$^+$-coupled HCO$_3^-$ transporters (NCBT) and isolated RNA from kidney and the Calu-3 cell line, which was derived from a metastatic site of a lung adenocarcinoma, as positive controls. The use of specific primers (*Supplementary file 1*) for each NCBT revealed that *SLC4A4*, *SLC4A5*, *SLC4A7*, and *SLC4A8* are expressed at mRNA levels in primary hAECs from three different individuals (P1, P2, P3, *Figure 1—figure supplement 1A-E*). *SLC4A10* showed near undetectable level of mRNA expression (*Figure 1—figure supplement 1F*). Interestingly, isoforms B/C of NBCe1 (known as the pancreatic isoform, *Figure 1—figure supplement 1A*) were more highly expressed than isoform A (known as the kidney isoform; *Figure 1—figure supplement 1E*). Data extraction from previously published RNA-seq results (*Saint-Criq et al., 2020*) (GEO series accession number GSE154905) confirmed the pattern of expression of these isoforms (*Figure 1—figure supplement 1G*) and revealed that *SLC4A4* is the most expressed member, closely followed by *SLC4A7*, *SLC4A8,* and *SLC4A5*. Other candidate HCO3- transporters such as Bestrophins were also expressed in these cells but with one to two logs lower than *SLC4A4*, *SLC4A7,* and *SLC4A8* (*Figure 1—figure supplement 1G*).

## SLC4A4 is central for bicarbonate secretion and intracellular pH homeostasis in human airway cells

We then tested whether there was an active NCBT under unstimulated and stimulated conditions in primary hAECs. The cell cultures were mounted in Ussing chambers in buffers containing either $HCO_3^-$ (but no $Cl^-$) or HEPES (no $HCO_3^-$), and treated basolaterally with the inhibitor S0859 (30 μM) followed by Forskolin (Fsk; 10 μM). Results, shown in *Figure 1* confirmed that, in the absence of stimulants, $HCO_3^-$ secretion was inhibited by S0859 (*Figure 1A and B*) and that this pharmacological inhibitor did not have any effect on short-circuit current (Isc) in the absence of $HCO_3^-$ (*Figure 1D and E*). Interestingly, S0859 did not reduce Fsk-stimulated $HCO_3^-$ secretion (*Figure 1C*) under these conditions. These results show that there is an electrogenic $HCO_3^-$ transporter at the basolateral membrane of hAECs, which is consistent with SLC4A4, since SLC4A7 and SLC4A8 are electroneutral (*Parker and Boron, 2013*), and SLC4A5 has been shown to be localized to the apical membrane of renal epithelial cells and cholangiocytes (*Gildea et al., 2015*; *Abuladze et al., 1998*). Next, we used intracellular pH ($pH_i$) measurements to functionally investigate NBCe1 activity using a $CO_2$-induced acidification protocol (*Theparambil et al., 2015*). As shown in *Figure 1E*, exposing cells bilaterally to a $HCO_3^-/CO_2$-gassed KRB solution induced a transient acidification. On the other hand, an apical-only $CO_2$ exposure, in the absence of basolateral $HCO_3^-$, induced a sustained acidification (of the same amplitude as with bilateral $HCO_3^-/CO_2$, *Figure 1F and H*), that recovered when $HCO_3^-$ was re-introduced basolaterally (*Figure 1F,I*). This $pH_i$ recovery depended on the presence of $Na^+$ in the basolateral solution (*Figure 1G*) consistent with a $Na^+$-coupled $HCO_3^-$ transporter, which was confirmed using S0859 which blocked the $pH_i$ recovery from the $CO_2$-induced acidification. In order to isolate NBCe1-dependent changes in $pH_i$, the contribution of $Na^+/H^+$ exchanger NHE was inhibited using 100 μM Dimethyl amiloride (DMA). In this condition, S0859 still significantly decreased the rate of $pH_i$ recovery from the $CO_2$-induced acidification (*Figure 1J and K*).

## Basolateral $HCO_3^-$ uptake is essential for ASL pH homeostasis

As Ussing chamber and $pH_i$ experiments use standard buffer solutions that are unlikely to fully resemble that of the ASL, we then used fully differentiated hAECs at the air-liquid interface to establish whether SLC4A4 is involved in airway epithelial pH homeostasis in a setting that is more physiologically relevant. In order to test whether $HCO_3^-$ transport by basolateral SLC4A4 impacted apical $HCO_3^-$ secretion under thin film conditions, we measured the effect of S0859 on ASL pH. First, S0859 was added basolaterally to primary hAECs and ASL pH continuously measured. NBCe1 inhibition significantly decreased ASL pH under resting conditions with a $t_{1/2}$ of 46 min (*Figure 2A,B*), and partially prevented the Fsk-induced increase in ASL pH which we have previously shown was via CFTR (*Delpiano et al., 2018*; *Saint-Criq et al., 2019*; *Figure 2C,D*), increasing the half-time response to Fsk from 25 to 31 min. Moreover, when S0859 was added after Fsk, it significantly reduced the Fsk-induced, CFTR-dependent, increase in ASL pH with a half-time of 14 min (*Figure 2E,F*) confirming the central role of SLC4A4 cotransporter in ASL pH homeostasis under both resting and stimulated conditions. It is worth noting that the changes in ASL pH were much slower than in the $pH_i$ experiments, which can be explained by the differences in technical (non-perfused ASL pH versus bilaterally perfused $pH_i$) and experimental conditions set-ups (chemical induced versus. $CO_2$-induced changes in pH).

Finally, immunolocalization of SLC4A4 protein in human airway tissues showed intracellular and basolateral membrane staining in epithelial cells that were also positively stained for acetylated-tubulin indicating that SLC4A4 is preferentially expressed in ciliated cells in human airways (*Figure 2G,H*).

## Bicarbonate secretion is calcium-activated in mouse trachea

To investigate the expression of SLC4 exchangers in mouse airway epithelium, we performed RT-PCR of epithelial cells from mouse tracheas and observed that several members of the SLC4 family including *Slc4a4*, *Slc4a5*, *Slc4a7*, and *Slc4a10* were expressed (*Figure 3—figure supplement 1* A-E). Studies of *Slc4a4* isoforms demonstrated that isoform B/C but not isoform A was expressed in mouse airways (*Figure 3—figure supplement 1F*). Next we characterized $HCO_3^-$ secretion in the mouse. Ussing chamber experiments performed in freshly excised mouse trachea using $HCO_3^-$ (*Figure 3A*) or HEPES (*Figure 3B*) buffered solutions, showed that UTP-induced an anion current that was significantly reduced in the absence of $HCO_3^-$ ($-138\pm28$ to $-82\pm15$ μA cm$^{-2}$; $p<0.01$ Mann-Whitney test), but no significant effect on the cAMP-induced anion secretion, or the amiloride-sensitive sodium

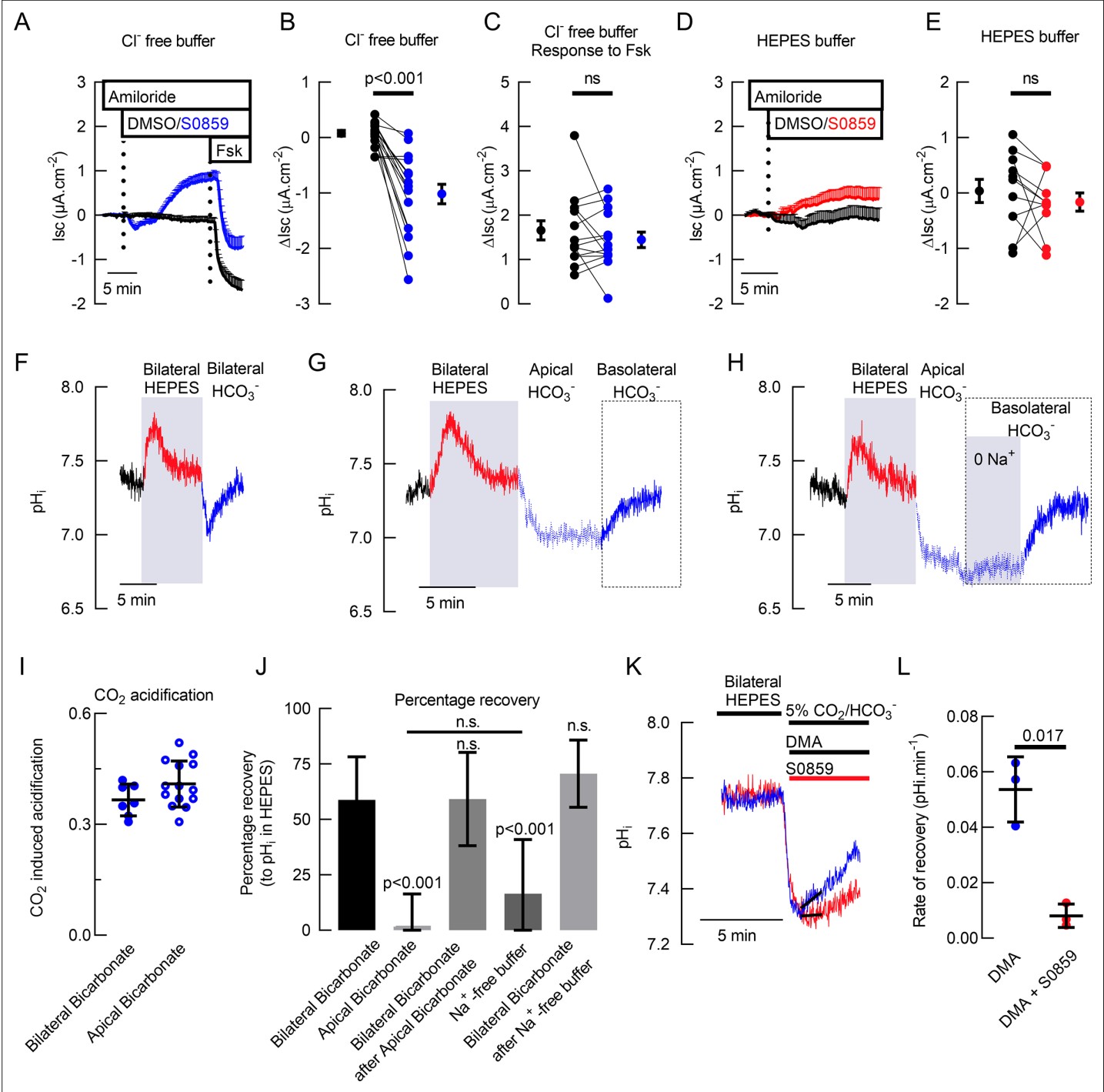

**Figure 1.** Basal bicarbonate secretion requires SLC4A4 activity in primary hAECs. Mean traces (+ standard error of the mean) of $I_{sc}$ in (**A**) $Cl^-$ free solution (n=14) and (**D**) $HCO_3^-$ free solution (n=10) of primary hAECs cultures (dotted lines represent the time of addition of drugs). Summary of S0859-induced changes in $I_{sc}$ in the presence (**B**) or absence (**E**) of $HCO_3^-$ in primary hAECs cultures and Fsk-induced $HCO_3^-$ secretion (**C**) in $Cl^-$ free buffer (on panels B, C,, and E, each dot represents an independent experiment; means ± sem are shown next to the individual data; respectively n=17, n=14, and n=11 independent experiments using cells from N=3 donors; two-tailed, paired t-test). (**F**) Representative $pH_i$ trace of primary hAECs bathed first in bilateral $HCO_3^-$ KRB (gassed with 5% $CO_2$) then bilateral Hepes buffered KRB (no $HCO_3^-$, no $CO_2$) and then bilateral $HCO_3^-$ KRB (gassed with 5% $CO_2$). $CO_2$ removal and re-introduction is marked by a transient increase and decrease in $pH_i$ respectively. (**G**) Representative trace of $pH_i$ recovery after $CO_2$-induced acidification in the absence of basolateral $HCO_3^-$. (**H**) Representative trace of $pH_i$ recovery after $CO_2$-induced acidification in the absence of basolateral $Na^+$ and $HCO_3^-$. (**I**) Summary of the magnitude of $CO_2$-induced acidification (bilateral bicarbonate, n=7, apical bicarbonate n=14, unpaired t-test, bars represent mean ± standard deviation (S.D.)). (**J**) Mean percentage of $pH_i$ recovery after perfusion of the different solutions (Bilateral

*Figure 1 continued on next page*

Figure 1 continued

Bicarbonate, n=7; Apical Bicarbonate, n=14; Bilateral Bicarbonate after Apical Bicarbonate⁻, n=5; Na⁺-free buffer, n=8; Bilateral Bicarbonate after Na⁺-free buffer, n=8; One-way ANOVA with Holm-Sidak correction for multiple comparison tests, bar graph represents mean ± S.D.). (**K**) Representative trace of intracellular pH measurements showing recovery from $CO_2$-induced acidification in the presence (red line) or absence (blue line) of NBC inhibitor S0859. (**L**) Summary of rates of recovery from $CO_2$-induced acidification in the presence of NHE inhibitor (Dimethyl Amiloride, DMA, 100 µM) and in the presence (red bar) or absence (blue bar) of S0859 (30 µM), (n=3, paired t-test; bars represent mean ± S.D.).

The online version of this article includes the following figure supplement(s) for figure 1:

**Figure supplement 1.** Expression of SLC4A family members of bicarbonate transporters in differentiated primary human airway epithelial cells.

absorption (*Figure 3C*) was detected as previously shown (*Anagnostopoulou et al., 2012*). Complementary studies in $HCO_3^-$ buffer showed that de novo synthesis of $HCO_3^-$ was not participating in the UTP-induced electrogenic anion secretion, as incubation of tracheas with acetazolamide didn't affect the magnitude of the UTP-induced current (–124±14 µA cm⁻²; p>0.05 One-way ANOVA).

Using the SLC4A4 blocker S0859, we observed the inhibition of the cAMP-induced anion current (ΔIsc –12.2±2.4 µA cm⁻²) suggesting that SLC4A4 might participate in the cAMP-response (*Figure 3—figure supplement 2A-C*). Nevertheless, when the TMEM16A/CFTR inhibitor, CaCCinhA01, was used to block the cAMP-induced current, further addition of S0859 was still able to induce a reduction in the current and of similar magnitude as shown in *Figure 3—figure supplement 2A* (*Figure 3—figure supplement 2D-F*; –10.1±2.8 µA cm⁻²), confirming that electrogenic-bicarbonate secretion was not significantly stimulated by cAMP, indicating that basal $HCO_3^-$ secretion occurs in mouse trachea. To confirm this last hypothesis we added S0589 to tissues pre-incubated with amiloride and observed a reduction in the basal current only in $HCO_3^-$ buffer (–13.9±3.3 to -3.0±1.6 Δ µA cm⁻² for $HCO_3^-$ vs HEPES buffer; p<0.02; Mann-Whitney; *Figure 3D–F*). The magnitude of basal $HCO_3^-$ secretion inhibited by S0859 was similar to the experiments summarized in *Figure 3—figure supplement 2C and F*. Of note, the addition of S0859 to the tracheas induced a fast and transient negative change in $I_{sc}$ as observed in *Figure 3D and E* and *Figure 3—figure supplement 2A and D*, that has been also observed in human cells (*Gorrieri et al., 2016*), and that might be due to off-targets of the blocker like other SLC4A or SLC16A transporters as previously described (*Heidtmann et al., 2015*; *Schwab et al., 2005*; *Ch'en et al., 2008*).

To further characterize the $Ca^{2+}$-activated anion secretion, the UTP response was tested with no involvement of cAMP-induced secretion. As can be observed (*Figure 3G and H*) the UTP-induced anion secretion in tracheas maintained in $HCO_3^-$ buffer was significantly reduced by previous addition of S0859 (–368±25 to -200±17 µA cm⁻²; p<0.001 One-way ANOVA). The UTP response was also reduced when $HCO_3^-$ was replaced with HEPES buffer (*Figure 3I*) (–199±25 µA cm⁻²; p<0.001 One-way ANOVA), but the addition of S0859 induced no significant reduction of the UTP-induced anion secretion in tissues maintained in HEPES buffer (*Figure 3J*) (–142±12 µA cm⁻²; p>0.05 One-way ANOVA).

## SLC4A4 participates in intracellular pH homeostasis in mouse airway epithelial cells

We reasoned that the UTP-induced $HCO_3^-$ exit would lead to cytoplasmic acidification and therefore we monitored intracellular pH of BCECF-loaded murine airway cells. As shown in *Figure 3L*, UTP induced an intracellular acidification (ΔpH_i –0.25±0.02) that was significantly reduced when cells were placed in low Cl⁻ buffer (ΔpH_i –0.11±0.02), indicating the existence of Cl⁻/$HCO_3^-$ exchange. *Figure 3M* summarizes changes in intracellular pH and includes experiments in HEPES buffer, which shows that UTP was almost unable to induce intracellular acidification (ΔpHi –0.01±0.01) in absence of $HCO_3^-$. To test if the UTP-induced intracellular acidification was dependent on SLC4A4 activity, we tested the S0859 inhibitor and observed acidification of the intracellular compartment and prevention of UTP-induced acidification (*Figure 3—figure supplement 2G-H*). Washout of S0859 partially restored pH_i and UTP-induced acidification. (*Figure 3—figure supplement 2G-H*; –0.05±0.01 to -0.13±0.03 ΔpH_i, for UTP with S0559 and UTP post washout, respectively; p>0.002; Mann-Whitney). These data suggest that both basal and UTP-induced $HCO_3^-$ secretion are dependent on SLC4A4 activity in mouse airway epithelial cells.

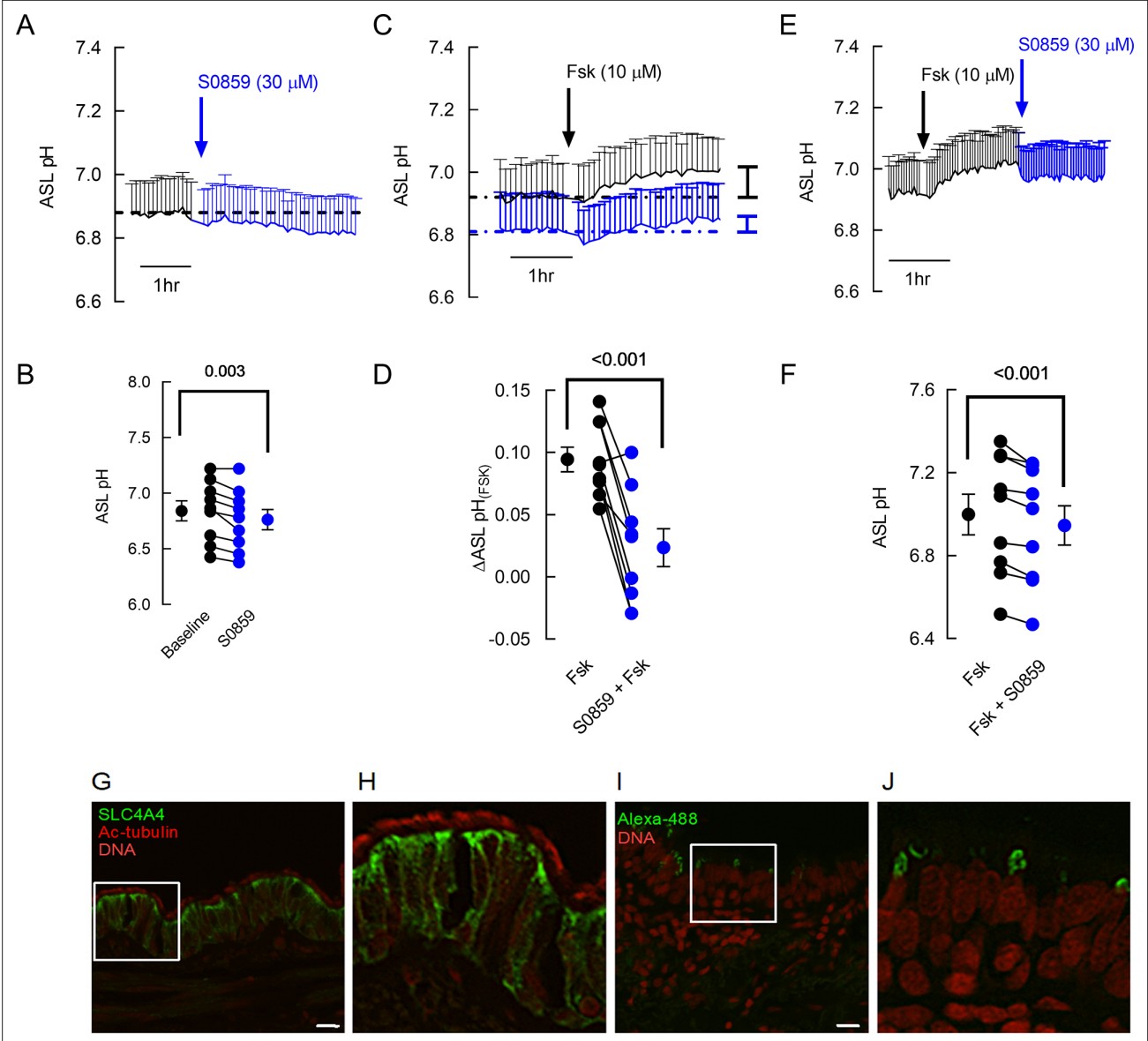

**Figure 2.** S0859 decreases resting ASL pH in primary human airway epithelial cells (hAECs), by blocking basolateral SLC4A4. (**A**) Mean (+ S.E.M.) trace of ASL pH recordings. ASL pH was measured under resting conditions for 1.5 hr before S0859 (30 µM) was added basolaterally. (**B**) Mean resting ASL pH before (black circles) and after (blue circles) addition of basolateral S0859 (n=9 independent experiments performed on epithelial cells from N=3 donors; paired t-test). (**C**) Mean (+ SEM) traces of ASL pH from hAECs pre-treated for 3 h with vehicle control (DMSO, black trace) or S0859 (30 µM, basolateral, blue trace). (**D**) Mean Forskolin (Fsk)-induced changes in ASL pH in hAECs treated with DMSO (black circles) or S0859 (blue circles) (n=9 independent experiments performed on epithelial cells from N=3 donors; paired t-test). (**E**) Mean (+ SEM) traces of ASL pH from hAECs treated with Fsk for 2.5 hr and then S0859. (**F**) Mean Fsk-stimulated ASL pH before (black circles) and after (blue circles) addition of basolateral S0859 (n=9 independent experiments performed on epithelial cells from N=3 donors; paired t-test). (**G–H**) SLC4A4 locates in the basolateral membrane of acetylated tubulin (Ac-tubulin) positive human airway epithelial cells. (**I–J**) correspond to negative controls for anti-SLC4A4 omitted antibody. Representative images of three different samples. Bar 20 µM.

## The genetic inactivation of *Slc4a4* induces a cystic fibrosis-like phenotype in mouse airways

As explained in the methods section, we decided to work with wild type and *Slc4a4*[-/-] on the hybrid background, at 16–20 days of age. First, we observed that *Slc4a4*[-/-] animals were affected by defects in tracheal cartilage formation with the presence of ventral gaps and abnormal patterns on the rostrocaudal side (*Figure 4A*). In wild-type animals, immunolocalization of SLC4A4 showed strong

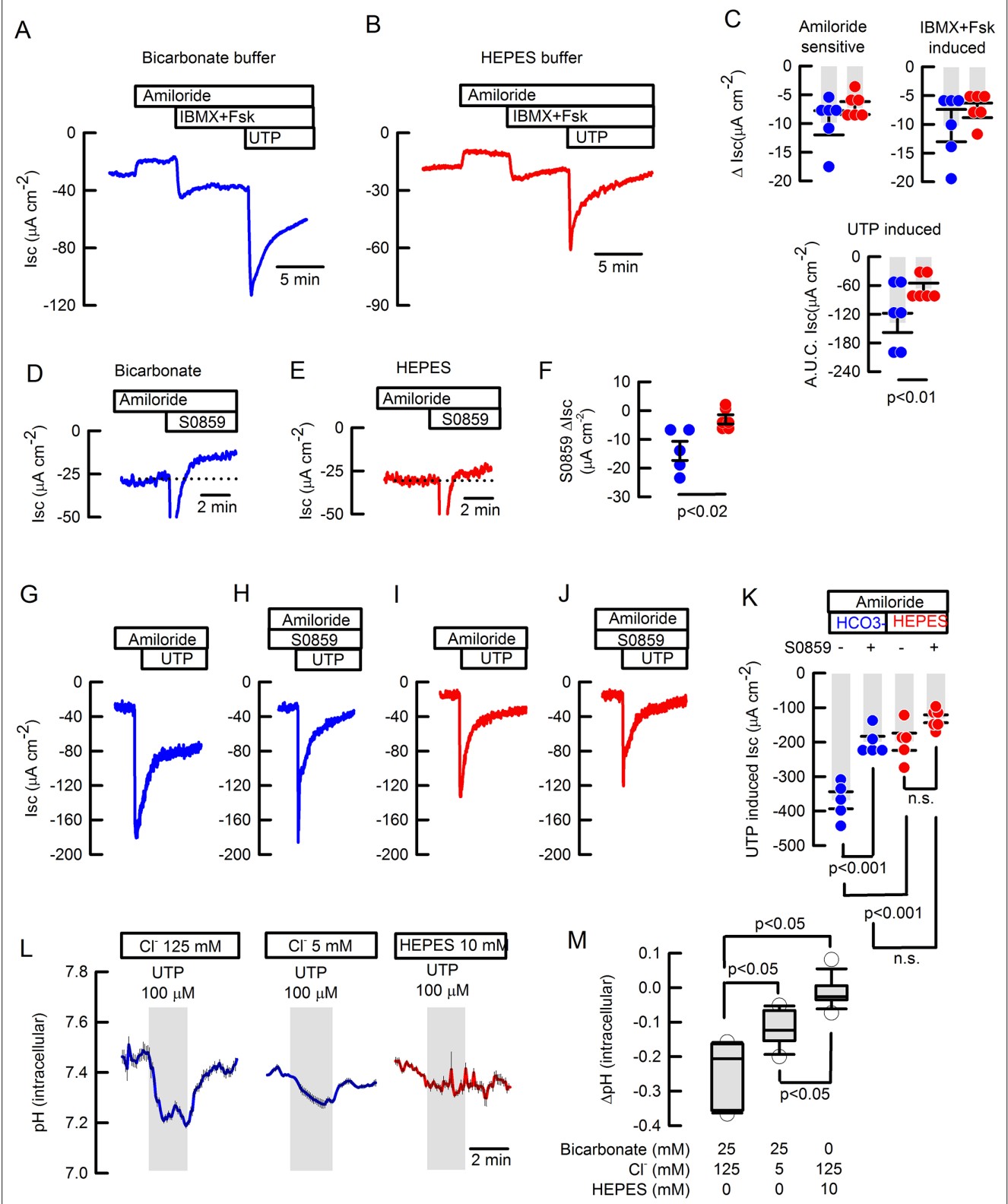

**Figure 3.** Basal and inducible bicarbonate secretion relies in SLC4A4 activity in mouse tracheal epithelium. Representative traces of $I_{SC}$ in
(**A**) bicarbonate and (**B**) HEPES buffer of freshly excised mouse tracheas. (**C**) Summary of $\Delta I_{SC}$ values for amiloride-sensitive $Na^+$ absorption, IBMX
+Fsk-induced and A.U.C. $I_{SC}$ UTP-induced anion secretion in mouse tracheas; n=6 for each condition; Mann-Whitney Rank Sum Test. Representative
$I_{SC}$ traces of S0859 effect on unstimulated tracheas in (**D**) bicarbonate and (**E**) HEPES buffer. (**F**) Summary of $\Delta I_{SC}$ values for S0859 effect, n=5 for each

*Figure 3 continued on next page*

*Figure 3 continued*

condition; Mann-Whitney Rank Sum Test. Representative $I_{SC}$ traces for UTP-induced anion secretion in absence (**G and I**) or presence (**H and J**) of 30 μM S0859 in buffer bicarbonate (**G and H**) or HEPES (**I and J**). All in the presence of 10 μM amiloride. (**K**) Summary of experiments as presented in G-J; n=5 but HEPES + S0859 n=6; ANOVA on Ranks. Bars are mean ± S.E.M. (**L**) Average response of determination of $pH_i$ in epithelial cells isolated from wild type mouse trachea and loaded with BCECF that were stimulated with 100 μM UTP and switched to 12.5 mM bicarbonate buffered solution. Experiments performed in bicarbonate buffer in blue and HEPES in red. (**M**) Summary of $\Delta pHi$ from experiments as those in (**L**) including a data set of cells maintained in HEPES buffer; Middle line of the box plot indicates the median; n=15 cells from four different mice, n=11 cells, three different mice and n=13 cells, three different mice respectively; ANOVA on Ranks.

The online version of this article includes the following figure supplement(s) for figure 3:

**Figure supplement 1.** Expression of SLC4 family members of bicarbonate transporters in mouse airway epithelial cells.

**Figure supplement 2.** Inhibition of SLC4A4 by S0859 reduces anion secretion and intracellular pH in mouse tracheal epithelium.

localization in the airway epithelium but the signal was nearly absent in the airways from the $Slc4a4^{-/-}$ mice (*Figure 4B*). Using the same antibody, we showed that SLC4A4 was preferentially expressed in CCSP-positive cells that correspond to Club cells and was excluded from cells positive to acetylated-Tubulin, that identify ciliated cells (*Figure 4C*). This pattern of expression was maintained in distal airway bronchi and bronchioles (*Figure 4—figure supplement 1C*). Further histological examination of the $Slc4a4^{-/-}$ mouse airways demonstrated the presence of adherent mucus at the surface of the tracheal epithelium (*Figure 4D* and *Figure 4—figure supplement 1C,D*) as well as in the bronchi (*Figure 4—figure supplement 1E-G*) and bronchioli (*Figure 4—figure supplement 1H*). Signs of damaged epithelium was also observed as interruptions in the epithelial layer facing the lumen in the $Slc4a4^{-/-}$ airways (*Figure 4D* and *Figure 4—figure supplement 1F,H*), that might explain the decreased $R_{te}$ of the tracheas placed in Ussing chambers and that prevented electrophysiological measurements in the $Slc4a4^{-/-}$ tracheas (*Figure 4E*).

## Genetic silencing of *Slc4a4* impairs intracellular pH homeostasis and mucociliary clearance in mouse airways

In order to validate the role of SLC4A4 in pH homeostasis of murine airways, UTP-induced intracellular acidification was studied in airway cells isolated from the $Slc4a4^{-/-}$ mice. We observed a decrease in the magnitude of intracellular acidification in $Slc4a4^{-/-}$ cells when compared to those from wild type animals during UTP stimulation (–0.24±0.01 to -0.14±0.01 $\Delta pH_{[i]}$, for wild type vs $Slc4a4^{-/-}$) suggesting that an important amount of $HCO_3^-$ accumulates in airways cells via SLC4A4 (*Figure 4F–G*). We also noticed that after UTP wash-out, the acidification persisted in the wild type cells but not in the $Slc4a4^{-/-}$ (*Figure 4—figure supplement 1J*; –0.09±0.02 to -0.02±0.01 $\Delta pH_{[i]}$, for wild type vs $Slc4a4^{-/-}$), suggesting that $HCO_3^-$ secretion was sustained by SLC4A4 activity. Clearance of plastic beads, as a way to measure MCC, was studied in freshly isolated mouse tracheas whose mucosal side was exposed to air. As shown in the polar plots in *Figure 4H* the plastic beads covered a larger distance in wild type tissues and, in some cases, retrograde movement of beads was observed in the $Slc4a4^{-/-}$ trachea. A similar reduction in distance travelled was observed in wild type tissues bathed in HEPES buffer. The speed of plastic bead movement and total distance covered are summarized in *Figure 4I and H*. The use of HEPES buffer in the $Slc4a4^{-/-}$ tracheas showed no further effect on both speed and total distance travelled by the beads. This data set demonstrates that mucociliary clearance is significantly decreased when $HCO_3^-$ transport is impaired after *Slc4a4* silencing.

## Discussion
### SLC4A4 is a critical component of the bicarbonate secretory machinery

In this study, we have established that the $Na^+$-coupled $HCO_3^-$ transporter SLC4A4 or NBCe1 is central for bicarbonate transport, coupling to apical proteins to efficiently deliver bicarbonate in human and mouse airways. As the specificity of S0859 for SLC4A4 has been discussed and is still unclear (*Heidtmann et al., 2015*; *Schwab et al., 2005*; *Ch'en et al., 2008*), it is uncertain from our data whether SLC4A4 is the main and only actor regulating bicarbonate uptake in primary hAECs. However its importance in airway pH homeostasis is supported by strong evidence from the use of knock-out animals. Early characterization in canine and human airway epithelium indicated that $HCO_3^-$ secretion

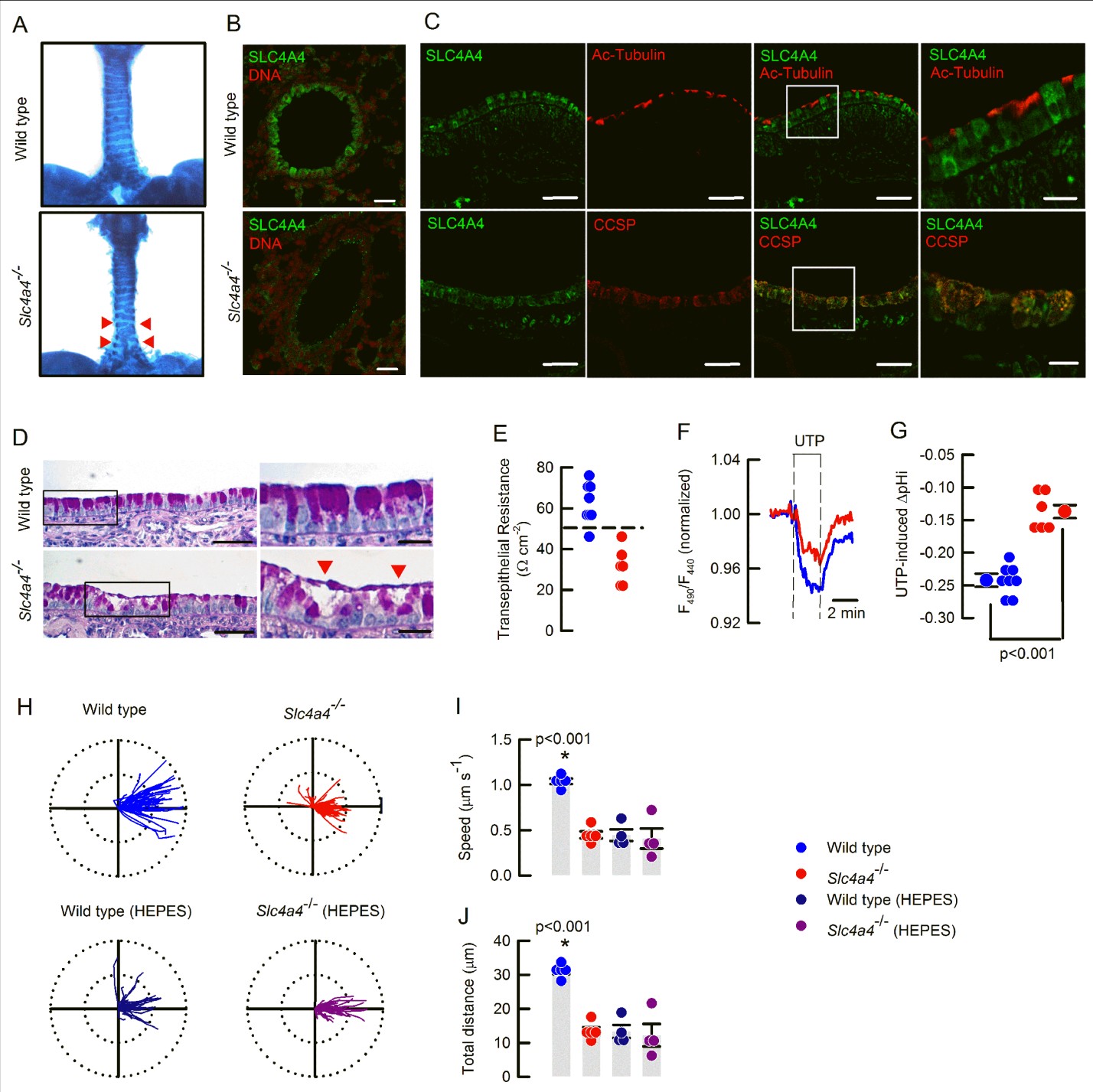

**Figure 4.** The *Slc4a4*[-/-] bear signs of muco-obstructive airway disease. (**A**) Ventral view of Alcian blue stained tracheas of 17 days old mice. Red arrow heads show incomplete cartilage rings in the *Slc4a4*[-/-] mouse; representative image of 3 animals per genotype. (**B**) SLC4A4 staining of epithelial cells is absent in the *Scl4a4*[-/-] lung tissues; representative images of three different animals per genotype. (**C**) Epithelial SLC4A4 co-localizes with CCSP; representative images of three different animals. (**D**) Mucin staining showing mucus adhered to the epithelial surface and epithelial damage (red arrow heads); representative images of five animals per genotype. (**E**) Summary of $R_{te}$ values for wild type and *Scl4a4*[-/-] tracheas; dashed line indicates 50 $\Omega$cm⁻². n=8 for wild types and n=6 for *Scl4a4*[-/-] tracheas. (**F**) Representative traces of $pH_i$ in airway cells from wild type (blue) and *Slc4a4*[-/-] (red) animals and (**G**) summary of UTP-induced $\Delta pH_i$ including mean ± S.E.M. and individual cells; n=8 cells for wild type and n=6 cells for *Scl4a4*[-/-], three different animals per genotype; Mann-Whitney Rank Sum Test. (**H**) Beads tracking of MCC experiments for wild type and *Slc4a4*[-/-] tracheas bathed with basolateral bicarbonate or HEPES buffer. Radius of the polar plots is 50 µm. Summary of MCC experiments for (**I**) speed, (**J**) total distance covered by beads. Bars indicate mean ± S.E.M.; n=5 for each genotype in bicarbonate buffer and n=4 for each genotype in HEPES buffer; ANOVA on ranks.

*Figure 4 continued on next page*

*Figure 4 continued*

The online version of this article includes the following figure supplement(s) for figure 4:

**Figure supplement 1.** *Slc4a4* silencing produces early lethality, weight loss, mucus accumulation and reduced intracellular pH.

occurs via CFTR, is activated by cAMP and dependent on basolateral Na$^+$ *Smith and Welsh, 1992*, similar features were observed in the Calu-3 cell line (*Devor et al., 1999*). Our experiments in hAECs corroborated the cAMP-activation and Na$^+$-dependence, that has also been observed by others in the same (*Kim et al., 2019*; *Gorrieri et al., 2016*) or other cell types (*Kurtz, 2014*). Nevertheless, and in contrast to human airway epithelium, our present data shows that HCO$_3^-$ secretion in the mouse airways is mostly Ca$^{+2}$- rather than cAMP-activated, but both human and mouse share a basal secretory component for HCO$_3^-$ previously observed in human bronchioles (*Shamsuddin and Quinton, 2019*), and that we now show is dependent on SLC4A4 activity.

Basolateral localization of SLC4A4 was previously demonstrated in the Calu-3 cell line (*Kreindler et al., 2006*) but up to the date there has been no other studies investigating SLC4A4 localization in native airway tissues. Here, we demonstrate that human airways express SLC4A4 in ciliated cells preferentially, and that the observed basolateral localization correlates with the functional evidence provided, reinforcing the role SLC4A4 in basolateral HCO$_3^-$ transport into the cells. Nevertheless, it must be considered that the sole expression of SLC4A4 does not assure HCO$_3^-$ secretion as compelling evidence has demonstrated that apical CFTR is also necessary. While the calcium-activated TMEM16A channel has been shown to be expressed in goblet cells (*Scudieri et al., 2012*), both HCO$_3^-$ transporters, CFTR and pendrin, have been localized in the apical membrane of ciliated hAECs cells supporting our observation (*Kim et al., 2019*; *Kreda et al., 2005*). Moreover, it has been demonstrated that SLC4A4-B, that we detected in human and mouse, and not the isoform A, is functionally coupled to CFTR through the IRBIT protein in pancreatic ducts (*Yang et al., 2009*; *Shirakabe et al., 2006*), further supporting the idea that a functional coupling in the same cell type is necessary for efficient HCO$_3^-$ secretion. Even though, CFTR and Pendrin expression increases in secretory cells after cytokine stimulation, SLC4A4 remained unaltered, suggesting that other basolateral HCO3- transporter support HCO3- secretion in human secretory cells during the inflammatory response (*Rehman et al., 2020*). Finally, CFTR distribution in airway epithelium has been under renewed scrutiny after the recent discovery of the pulmonary ionocyte, a rare airway epithelial cell type that contains the highest amount of *CFTR/Cftr* transcripts in human and mouse (*Montoro et al., 2018*; *Plasschaert et al., 2018*). Nevertheless, recent and detailed studies demonstrate that most human airway epithelial cell types express CFTR including ciliated cells as was initially demonstrated (*Engelhardt et al., 1994*; *Okuda et al., 2021*).

We noticed that SLC4A4 localization in mouse was different from human airway epithelium, as the mouse SLC4A4 was expressed in CCSP +and not ciliated cells. Previously, CFTR and TMEM16A channels were specifically located in non-ciliated cells of mouse airways (*Hahn et al., 2018*), a distribution also conserved in the rat (*Hahn et al., 2017*). Pendrin, was also detected in non-ciliated secretory MUC5AC + cells, suggesting that a functional coupling for HCO$_3^-$ secretion occurs in secretory non-ciliated cells of the mouse airways (*Jia et al., 2016*). Such a difference in expression of transporter proteins among human and mouse airways has frequently been described; for example CFTR, is not the principal Cl$^-$ transporter in the mouse airways and consequently silencing or mutation of *Cftr* in mice did not produce CF disease of the lungs (*Ratcliff et al., 1993*; *Snouwaert et al., 1992*). Furthermore, the expression in human and not mouse airways of another protein that influence ASL pH, the ATP12A H$^+$/K$^+$ ATPase, magnifies ASL acidification in human airways in CF disease (*Shah et al., 2016*).

## Impaired bicarbonate secretion produces a CF-like phenotype in the mouse airways

We demonstrated that SLC4A4 activity is pivotal for HCO$_3^-$ secretion. Inhibition of SLC4A4 induced acidification of the ASL in hAECs, an observation that suggests altering HCO$_3^-$ delivery can initiate a CF-like phenotype, and that was confirmed in the *Slc4a4*$^{-/-}$ mouse model. Previous observations obtained by inducing silencing of *Tmem16a* in the mouse, or using cells carrying natural mutations in CFTR, affected both Cl$^-$ and HCO$_3^-$ secretion making it difficult to understand what the consequences of reduced transport for each anion alone are (*Gorrieri et al., 2016*; *Rock et al., 2009*).

As observed here, the silencing of *Slc1a4* induced a muco-obstructive phenotype in the mouse, whereas the silencing of *Slc12a2*, which encodes the bumetanide-sensitive NKCC1 co-transporter, which is essential for Cl⁻ accumulation and secretion, did not (*Grubb et al., 2001*), suggesting that lack of $HCO_3^-$ is pathologically more relevant than Cl⁻ secretion in mouse airways. Indeed, a significant amount of experimental evidence, using different models and species, is consistent with our findings in the *Slc4a4⁻/⁻* mouse. For example, hAEC from CF patients produce acidic ASL and mucus that is more viscous than in cells from non-CF donors (*Tang et al., 2016*), the acidification of the ASL induced abnormal epithelial immune responses and reduced MCC that could be reversed after $HCO_3^-$ supplementation (*Pezzulo et al., 2012*; *Shah et al., 2016*; *Ferrera et al., 2021*; *Simonin et al., 2019*; *Birket et al., 2018*; *Cooper et al., 2013*). Furthermore, $HCO_3^-$ secretion is necessary for proper mucus release from hAECs (*Gorrieri et al., 2016*), and maintenance of normal amounts of ENaC-mediated Na⁺ absorption and ASL volume (*Garland et al., 2013*), functions that become abnormal due to acidic ASL pH. Even though technically challenging issues prevented us from measuring ASL pH in the mouse trachea, the reduction of $HCO_3^-$ transport demonstrated here after SLC4A4 inhibition is consistent with the expected muco-obstructive phenotype, including reduced MCC and mucus accumulation.

## Extra-renal phenotypes after *Slc4a4* inactivation and human mutations

Mouse models of *Slc4a4* silencing have shown differences in phenotypes depending on the isoforms affected. The *Slc4a4*-null animal, corresponding to the one used in the present studies, and the *Slc4a4*^W516X/W516X avatar mouse engineered to mimic a non-sense mutation found in a human patient, are affected by severe metabolic acidosis due to proximal renal tubular acidosis (pRTA; *Gawenis et al., 2007*; *Lo et al., 2011*). Even though, decreased plasma $HCO_3^-$ is observed in the *Slc4a4⁻/⁻* mice (5.3±0.5 mM; *Gawenis et al., 2007*), is unlikely that reduced $HCO_3^-$ availability influenced mucus accumulation in the airways, as we have previously determined that the transporter is fully saturated at 3 mM $HCO_3^-$ (*Theparambil et al., 2014*).

Extra-renal manifestations of *Slc4a4* silencing in the mouse include growth retardation, ocular band keratopathy, splenomegaly, abnormal dentition and intestinal obstructions, most of which mimics the clinical findings observed in human patients. Nevertheless lung defects have not been reported in human patients, and observation which could be explained by the fact that human disease is milder than in mouse models. This might be related to the fact that at birth human kidneys are functionally more mature than in mouse (*Takahashi et al., 2000*) and for example, while heterozygous animals present mild pRTA, human patients bearing heterozygous SLC4A4 mutations did not show any signs of disease (*Gawenis et al., 2007*; *Lo et al., 2011*). Even though, a compensatory activity of other Na⁺-coupled $HCO_3^-$ cotransporters, or exchangers, has been discarded in mouse, this possibility has not been examined in humans where compensatory activity might benefit patient's health (*Lo et al., 2011*).

Both the *Slc4a4⁻/⁻* and W516X-mutant animals die soon after weaning as we also observed, but while the specific knock-out mouse for SLC4A4 isoform A (NBCe1-A) has a normal life span, the knock-out mouse for SLC4A4 isoforms B and C (NBCe1-B/C) is still lethal. This suggests that the cause of death of the animals was not due to metabolic acidosis, but rather due to extra renal phenotypes worsened by pRTA (*Salerno et al., 2019*; *Lee et al., 2018*). This observation might also be linked to the influence of modifier genes as observed in human patients and CF mouse models (*Rozmahel et al., 1996*; *Cutting, 2010*; *Philp et al., 2018*). In this regard, our observation that lethality is dependent on the genetic background of the animals supports such a possibility.

It will be of interest to use the *Slc4a4* animal models and generate an airway specific null mouse to study the pathogenesis of muco-obstructive diseases of the lungs due to reduced $HCO_3^-$ secretion. We believe that the elucidation of the transport systems that participate in pH maintenance in the airways offers the chance of increasing our current knowledge of the impact of impaired bicarbonate transport during health and disease.

## Materials and methods

### In vitro human airway epithelial cells studies

#### Cell culture

Primary hAECs were a kind gift from Dr. Scott H. Randell (Marsico Lung Institute, The University of North Carolina at Chapel Hill, United States). The cells were obtained under protocol #03–1396 approved by the University of North Carolina at Chapel Hill Biomedical Institutional Review Board. Cells were expanded using the conditionally reprogrammed cell (CRC) culture method as previously described (*Suprynowicz et al., 2012*). Briefly, cells were seeded on 3T3J2 fibroblasts inactivated with mitomycin C (4 mg/ml, 2 hr, 37 °C) and grown in medium containing the ROCK inhibitor Y-27632 (10 mM, Tocris Biotechne, #1254) until they reached 80% confluence. Cells then underwent double trypsinization to remove the fibroblasts first and then detach the hAECs from the P150 dish. At that stage, cells were counted and could be frozen down. Cryopreserved cells were seeded onto semi-permeable supports (6.5 or 12 mm) in bilateral differentiating medium (ALI medium) as previously described (*Randell et al., 2011*). The apical medium was removed after 3–4 days and cells were then allowed to differentiate under air-liquid interface (ALI) conditions. Ciliogenesis started approximately 12–15 days after seeding and cells were used for experiments between days 25 and 35 after seeding.

#### Intracellular pH measurements

Primary airway epithelial cells, grown on 12 mm Transwell inserts, were loaded with the pH-sensitive, fluorescent dye BCECF-AM (10 µM, ThemoFisher Scientific #B-1150) for 1 hr in a Na-HEPES buffered solution (130 mM NaCl, 5 mM KCl, 1 mM $MgCl_2$, 1 mM $CaCl_2$, 10 mM Na-HEPES, and 5 mM D-glucose set to pH 7.4) at 37 °C. Cells were mounted on to the stage of a Nikon fluor inverted microscope and perfused with a modified Krebs (KRB) solution (115 mM NaCl, 5 mM KCl, 25 mM $NaHCO_3$, 1 mM $MgCl_2$, 1 mM $CaCl_2$, and 5 mM D-glucose) gassed with 5% (v/v) $CO_2$/95% (v/v) $O_2$ or with a Na-HEPES-buffered solution gassed with 100% $O_2$. Solutions were perfused across the apical and basolateral membranes at 37 °C at a speed of 3 ml min$^{-1}$ and 6 ml min$^{-1}$, respectively. To test the sodium dependence of bicarbonate transport, a $Na^+$-free KRB solution was used in which 115 mM NMDG-Cl replaced NaCl, and 25 mM choline-$HCO_3$ replaced $NaHCO_3$. To measure the effect of NBC inhibition on the recovery from $CO_2$ -induced acidification, epithelial cells were perfused basolaterally with 100 µM DMA (dimethyl amiloride, Sigma-Aldrich #A4562) to inhibit sodium-dependent hydrogen exchangers (NHEs) and 30 µM S0859 (Sigma-Aldrich #SML0638) to inhibit NBC. Intracellular pH ($pH_i$) was measured using a Life Sciences Microfluorimeter System in which cells were alternately excited at 490 and 440 nm wavelengths every 1.024 s with emitted light collected at 510 nm. The ratio of 490–440 nm emission was recorded using PhoCal 1.6 b software and calibrated to $pH_i$ using the high $K^+$/nigericin technique (*Turner et al., 2016*) in which cells were exposed to high $K^+$ solutions containing 10 µM nigericin, set to a desired pH, ranging from 6 to 7.5. For analysis of $pH_i$ measurements, $\Delta pH_i$ was determined by calculating the mean $pH_i$ over 60 s resulting from treatment. The initial rate of $pH_i$ change ($\Delta pHi/\Delta t$) was determined by performing a linear regression over a period of at least 40 s.

#### Short-circuit current measurements in human airway epithelial cells

Cells grown on 6.5 mm inserts were mounted into the EasyMount Ussing Chamber Systems (VCC MC8 Physiologic Instrument, tissue slider P2302T) and bathed in bilateral $Cl^-$ free $HCO_3^-$ KRB (25 mM $NaHCO_3$, 115 mM Nagluconate, 2.5 mM $K_2SO_4$, 6.0 mM Ca-gluconate, 1 mM Mg-gluconate, 5 mM D-glucose) continuously gassed and stirred with 5% (v/v) $CO_2$/95% (v/v) $O_2$ at 37 °C or in bilateral NaHEPES buffered solution continuously gassed and stirred with 100% $O_2$ at 37 °C. Monolayers were voltage-clamped to 0 mV and monitored for changes in short-circuit current ($\Delta I_{sc}$) using Ag/AgCl reference electrodes. The transepithelial short-circuit current ($I_{sc}$) and the Transepithelial electrical resistance ($R_{te}$, expressed in $\Omega$ cm$^2$) were recorded using Ag–AgCl electrodes in 3 M KCl agar bridges, as previously described (*Saint-Criq et al., 2013*), and the Acquire & Analyze software (Physiologic Instruments) was used to perform the analysis. Cells were left to equilibrate for a minimum of 10 min before amiloride (10 µM, apical, Sigma-Aldrich #A7410) and S0859 (30 µM, basolateral) were added. Results were normalized to an area of 1 cm$^2$ and expressed as Isc (µAmp.cm$^{-2}$). The number of replicates was determined using previously obtained short circuit current measurements.

## ASL pH measurements

ASL pH measurements were performed as previously described (*Delpiano et al., 2018*; *Saint-Criq et al., 2019*). Briefly, cells grown on 6.5 mm transwells were washed apically with modified Krebs solution for 15 min at 37 °C, 5% $CO_2$. The ASL was stained using 3 µl of a mixture of dextran-coupled pH-sensitive pHrodo Red (0.5 mg/ml, $\lambda$ ex: 565 nm, $\lambda$ em: 585 nm; ThermoFisher Scientific, #P10361) and Alexa Fluor 488 (0.5 mg/ml, $\lambda$ ex: 495 nm, $\lambda$ em: 519 nm; ThermoFisher Scientific #D-22910) diluted in glucose-free modified Krebs buffer, overnight at 37 °C, 5% $CO_2$. The next day, fluorescence was recorded, every 5 min, using a temperature and $CO_2$-controlled plate reader (TECAN SPARK 10 M) and forskolin (Tocris Biotechne #1099) and S0859 were added basolaterally at indicated times. The ratio of pHrodo to Alexa Fluor 488 was converted to pH using a calibration curve obtained by clamping apical and basolateral pH in situ using highly buffered solutions between 5.5 and 8 (*Delpiano et al., 2018*). To prevent inter-experiment variability, the standard curve calibration was performed on each independent experiment. Changes in ASL pH (ΔASLpH) were calculated by averaging five time points (average pH over 25 min) before and 2 hrs after the addition of the molecules (Fsk/S0859). The number of replicates was determined using previously obtained ASL pH data (*Saint-Criq et al., 2019*). Using Cohen's d, a power analysis showed that the sample size of 9 independent experiments has an 80% power to detect an effect size of 0.35 pH unit, assuming a 5% significance level and a two-sided test (baseline ASL pH = 6.82 ± 0.25).

## RNA extraction and PCR analysis

RNA isolation from cells was performed using PureLink RNA Mini Kit (Ambion, Life technologies, #12183018 A), following the manufacturer's instructions. Briefly, lysates were mixed with 70% ethanol and loaded onto a silica-membrane column. Columns were washed with different buffers and total RNA was eluted in DNAse and RNAse-free water and stored at –80 °C until use. DNase treatment was performed on 300 ng RNA prior to Reverse Transcription Polymerase Chain Reaction (RT-PCR) using RNAse-free DNAse I (Roche, # 04716728001) at 37 °C for 10 min. Reaction was then stopped by increasing the temperature to 70 °C for 10 min. Complementary DNA (cDNA) was synthesized from total RNA (300 ng) using M-MLV Reverse Transcriptase (Promega, #M1701) as per supplier's protocol (1 hr at 37 °C followed by 10 min at 70 °C). Expression of SLC4A family members was evaluated by PCR using specific primers (*Supplementary file 1*), in a total volume of 25 µL containing 2 µL cDNA template, 5 µL 5 X Q5 reaction buffer, 0.5 µL 10 mM dNTP, 1.25 µL 10 µM of each primer, 0.25 µL Q5 high Fidelity Polymerase (New England Biolabs Inc, M0491), 5 µL 5 X Q5 high enhancer (Denaturation, 98 °C, 30 sec; 98 °C 5 sec, 72 °C 30 sec –72 °C 20 sec) x35 cycles; Final Extension, 72 °C, 2 min. PCR products were loaded onto a SYBR Safe DNA stain (Life Technologies, cat. no. S33102)-containing, 2% agarose gels in TBE and electrophoresis was ran at 90 V for 1.5 hr. Amplified products were visualized using LAS-3000 Imaging System (Fuji).

## **Ex vivo murine airway studies**

### Animals

The *Slc4a4⁻/⁻* mice was obtained from its laboratory of origin (*Gawenis et al., 2007*) and bred in the original 129S6/SvEv/Black Swiss background or C57BL/6 J. As observed in *Figure 4—figure supplement 1A-B* animals maintained in the C57BL/6 J background were severely affected by weight loss and lethality before weaning (day 21 post birth). Therefore experiments were performed in the hybrid animals. The wild type C57BL/6 J mice were from The Jackson Laboratories (USA). Animals were bred and maintained in the Specific Pathogen Free mouse facility of Centro de Estudios Científicos (CECs) with access to food and water ad libitum. 8–12 weeks-old or 16–20 days-old, male or female mice were used. Unless otherwise stated, all procedures were performed after mice were deeply anesthetized via i.p. injection of 120 mg/kg ketamine and 16 mg/kg xylazine followed by exsanguination. All experimental procedures were approved by the Centro de Estudios Científicos (CECs) Institutional Animal Care and Use Committee (#2015–02) and are in accordance with relevant guidelines and regulations.

### Ussing chamber experiments

Tracheae were placed in P2306B of 0.057 cm² (*Figure 3A–K* and *Figure 4—figure supplement 1A-F*) or P2307 of 0.031 cm² (*Figure 4E*) tissue holders and placed in Ussing chambers (Physiologic

Instruments Inc, San Diego, CA, USA). Tissues were bathed with bicarbonate-buffered solution (pH 7.4) of the following composition (in mM): 120 NaCl, 25 NaHCO$_3$, 3.3 KH$_2$PO$_4$, 0.8 K$_2$HPO$_4$, 1.2 MgCl$_2$, 1.2 CaCl$_2$ or HEPES buffer: 130 NaCl, 5 KCl, 1 MgCl$_2$, 1 CaCl$_2$, 10 Na-HEPES (pH adjusted to 7.4 using HCl); supplemented with 10 D-Glucose, gassed with 5% CO$_2$–95% O$_2$ (bicarbonate buffer) or 100% O$_2$ (HEPES buffer) and kept at 37 °C. The transepithelial potential difference referred to the serosal side was measured using a VCC MC2 amplifier (Physiologic Instruments Inc). The short-circuit currents were calculated using the Ohm's law as previously described (*Vega et al., 2020*). Briefly electrogenic Na$^+$absorption was inhibited using 10 µM amiloride (Sigma #A7410), cAMP-dependent anion secretion was induced using an IBMX +Forskolin mixture of 100 µM IBMX (Sigma #I5879)+1 µM forskolin (Sigma #F6886), Ca$^{2+}$-dependent anion secretion was induced by 100 µM UTP (Sigma #U6750). Acetazolamide (100 µM) was used to inhibit bicarbonate production (Sigma #A7011), to block SLC4A4 30 µM S0859 was used (kindly donated by Juergen Puenter, Sanofi-Aventis, France and dissolved in ethanol), and 30 µM CACC$_{inh}$A01 (Calbiochem) to inhibit CFTR and TMEM16A channels. The $\Delta$I$_{sc}$ values were calculated by subtracting I$_{sc}$ values values before from I$_{sc}$ values after the addition of drugs but UTP induced current was calculated as the area under the curve (A.U.C.) of the first 5 min post UTP addition using the Acquire & Analyze 2.3 v software. Tissues with R$_{te}$ values below 50 $\Omega$cm$^2$ were discarded as they were not suitable for bona fide electrophysiological determinations (*Gianotti et al., 2016*).

## Airway cells isolation and intracellular pH determinations

Tracheae were incubated with Pronase 25 µg/ml at 37 °C for 30 min. Then the trachea was placed in a petri dish with DMEM-F12, and the airway epithelium was dissociated by scrapping with tweezers, the cells were collected and spun at 3000 r.p.m. for 5 min at room temperature and the supernatant was removed. The cell pellet was incubated with 500 µl of trypsin 1 X at 37 °C for 5 min and centrifuged at 3000 r.p.m for 5 min at room temperature and the supernatant was removed. The airway epithelial cells were resuspended into 150 µl of DMEM-F12 medium supplemented with 10% fetal bovine serum (FBS) and seeded on poly-L-lysine coated 25 mm glass coverslips in 35 mm Petri dishes. Freshly isolated cells from mouse trachea were loaded with 0.5 µM BCECF-AM (ThermoFisher Scientific #B-1170) for 10 min at 37 °C. After loading, cells were washed and incubated 30 min in imaging solution (see below) to allow probe de-esterification. Cells were mounted into an open chamber and superfused with a bicarbonate buffer imaging solution of the following composition (in mM): 120 NaCl, 25 NaHCO$_3$, 3.3 KH$_2$PO$_4$, 0.8 K$_2$HPO$_4$, 1.2 MgCl$_2$, 1.2 CaCl$_2$ and bubbled with air/5% CO$_2$. For experiments without bicarbonate, the same HEPES buffer as in Ussing chamber experiments was used and bubbled with 100% O$_2$. In the low chloride bicarbonate buffer, 120 mM NaCl was replaced by 115 mM Na-Gluconate and 5 mM NaCl. Experiments were carried out on an Olympus IX70 inverted microscope equipped with a 40 X oil-immersion objective (NA 1.3), a monochromator (Cairn, UK) and a CCD Hamamatsu Orca camera (Hamamatsu, Japan), controlled by Kinetics software. All solutions were superfused at 37 °C using an in-line heating system (Warner instruments). BCECF was excited sequentially at 490 nm and 440 nm for 0.05–0.1 s and emission collected at 535/15 nm. The F490/F440 ratio was computed and transformed to pH units by performing a pH-clamp. Briefly, cells were exposed to 5 µM nigericin and 20 µg/ml gramicidin in a buffer composed of (in mM) 10 HEPES, 129 KCl, 10 NaCl, 1.25 MgCl$_2$, 1 EGTA, 10 glucose, with pH values ranging between 6.8–7.8 and the observed changes in fluorescence were quantified and used to construct a calibration curve.

## Airway cell isolation and RT-PCR

Tracheae were incubated with Pronase 25 µg/ml at 37 °C for 30 minutes. Trachea was placed in DMEM-F12 with 10 mM D-glucose and epithelium was isolated by scrapping with tweezers and further homogenized in 250 µl Trizol (Trizol Reagent) and RNA isolated following the manufacturer´s instructions. The dried pellet of RNA was resuspended with 35 µl nuclease free water and stored at –80 °C. Genomic DNA was removed through DNase treatment. The concentration and integrity of RNA isolation was verified using a NanoDrop spectrophotometer Maestrogen. RNA was reverse transcribed into cDNA using the ImProm-IITM Reverse Transcription System (Promega) following manufacturer's recommendations. The specific primer pairs used for *Slc4a4*, *Slc4a5*, *Slc4a7*, *Slc4a8*, *Slc4a10*, *Slc4a4*-A, and *Slc4a4*-B PCR amplification are provided in *Supplementary file 2*. PCR amplification

was performed starting with 3-min template denaturation step at 95 °C, followed by 40 cycles of denaturation at 95 °C for 30 s and combined primer annealing/extension at temperature as appropriate.

## Histology and immunofluorescence

Human tissues were obtained from the Pathological Anatomy Subdepartment of Hospital Base Valdivia (Valdivia, Chile) and corresponded to surgical resections of lung tumors that contained normal parenchyma including epithelium. The studies were approved by the Comité Ético Científico of the Servicio de Salud Los Ríos (CEC-SSV 443–2021). To obtain mice tissues the animals were placed in a 1 litre induction chamber under 1000 ml min⁻¹ flow of air containing 2.5% isoflurane. Then kept under anaesthesia with 2% isoflurane at a constant flow rate of 500 ml min⁻¹ using a mask. Mice were euthanized by exsanguination by severing the inferior vena cava under deep anaesthesia. Mice were perfused with 4% paraformaldehyde (PFA). Trachea and lung were removed and incubated overnight in 4% PFA at 4 °C. Paraffin sections (4 μm) were treated with Trilogy 1 X (Cell Marque cat# 920 P-06), blocked with 2.5% normal goat serum (Vector Laboratories cat# S-1012), and incubated with 1:100 anti-NBCe1 (anti SLC4A4; Alomone cat#ANT-075) 4 °C overnight. Sections were incubated with secondary antibody 1:2000 anti-rabbit Alexa Fluor 488 (Invitrogen cat# A-11008) 2 hr at room temperature. For colocalization 1:100 anti-NBCe1 was incubated with 1:1000 anti-Clara Cell Secretory Protein (CCSP; Merck Millipore cat#07–623) or 1:200 anti- alpha tubulin (Santa Cruz cat#sc-5286) overnight at 4 °C, and incubated with secondary antibody 1:2,000 anti-rabbit Alexa Fluor 568 (Life Technologies cat#A-11011) or 1:2000 anti-mouse Alexa Fluor 568 (Life Technologies cat#A-11004) respectively. Nuclei were stained with 1:2000 propidium iodide (Invitrogen cat#P21493). All immunofluorescence images were captured using a confocal microscope (Olympus FV1000).

## Whole-mount trachea Alcian blue cartilage staining

Tissue was fixed in 95% ethanol overnight followed by 2 hr staining with 0.03% Alcian blue (Sigma cat#A5268) dissolved in 80% ethanol and 20% acetic acid. Samples were cleared in 2% KOH, and pictures taken under a stereomicroscope.

## Mucociliary clearance determination

Speed of polystyrene beads in trachea samples was determined as previously described (*Vega et al., 2020*). Briefly, the tracheas were isolated and mounted with insert needles onto extra thick blot paper (Bio-Rad) and transferred into a water-saturated chamber at 37 °C. The filter paper was perfused with $HCO_3^-$ buffered solution of the following composition (in mM): 120 NaCl, 25 NaHCO_3, 3.3 KH_2PO_4, 0.8 K_2HPO_4, 1.2 MgCl_2, 1.2 CaCl_2 (gassed with 5% (v/v) $CO_2$/95% (v/v) $O_2$ to maintain solution pH close to 7.4) at a rate of 1 ml min⁻¹ and at 37 °C. Polystyrene black dyed microspheres (diameter 6 μm, 2.6% solid-latex, Polybead, Polyscience Inc) were washed and resuspended in $HCO_3^-$ or HEPES solution and 4 μl of particle solution with 0.3% latex were added onto the mucosal surface of the trachea. Particle transport was visualized every 5 s for 15 min using a ZEISS SteREO Discovery.V12, with digital camera Motic (Moticam 5.0). Particle speed was calculated with NIH ImageJ software and speed of MCC was expressed in μm/s. To inhibit $HCO_3^-$ secretion, $HCO_3^-$ solution was substituted for the $HCO_3^-$-free solution (HEPES) of the following composition (in mM): 145 NaCl, 1.6 K_2HPO_4, 0.4 KH_2PO_4, 1.0 MgCl_2, 1.3 CaCl_2 throughout the experiments. In these experiments, the $HCO_3^-$ free solution (HEPES) was gassed with 100% $O_2$ gas to maintain solution pH close to 7.4. Beads tracked on each tissue were averaged and were used as corresponding to one tissue sample.

## Statistical analysis

Experiments in human cells were analysed using GraphPad Prism v9. Statistical analysis was performed taking into account the number n of independent repetitions (done in different days) of the experiments (stated in the Figure legends) using cells at different passage numbers from 3 different donors. Thus n numbers given in the figure legends are considered as biological replicates. Statistical tests used are indicated in the figure legends and p-values are shown in the figures. Experiments using animals were analysed using the Sigmaplot 12 software. Statistical analysis was performed considering as n the number of animals used as source for tissues. These n numbers given on figure legends are considered biological replicates. For $pH_i$ experiments isolated cells were from at least 3 different animals per group and statistical analysis was performed using values from individual cells. ANOVA

on Ranks was used for comparisons of more than 2 data sets while Rank Sum test for comparison of data with two data sets. For survival analysis Log Rank test was used. p-Value of <0.05 was considered statistically significant. Sample size for experiments in human cells were calculated using Cohen's d, a power analysis showed that the sample size of five independent experiments has a 90% power to detect a difference, assuming a 5% significance level and a two-sided test. Calculations were based in previously published data for intracellular pH (*Turner et al., 2016*), ASL pH measurements (*Saint-Criq et al., 2019*) and unpublished data set for short-circuit currents. Sample size for animal experiments was calculated using previous published data for Ussing chamber experiments (*Vega et al., 2020*; *Gianotti et al., 2016*), intracellular pH (*Theparambil and Deitmer, 2015*) and mucociliary clearance (*Vega et al., 2020*).

## Acknowledgements

This work was supported by two CF Trust Strategic Research Centre grants (SRC003 and SRC013) and a Medical Research Council (MRC) Confidence in Concept grant (MC_PC_15030) and FOND-ECYT 1221257 (CAF). The Centro de Estudios Científicos (CECs) was funded by the Base Financing Programme of CONICYT, Chile. Cells from Dr. Randell were supported by Cystic Fibrosis Foundation grant (BOUCHE15R0) and NIH grant (P30DK065988). We would like to acknowledge Drs. Scott H Randell and Leslie Fulcher (Marsico Lung Institute, The University of North Carolina at Chapel Hill, United States) for providing human primary airway epithelial cells from the UNC CF Center Tissue Procurement and Cell Culture Core, and Git Chung for providing human kidney RNA sample.

## Additional information

### Funding

| Funder | Grant reference number | Author |
| --- | --- | --- |
| Cystic Fibrosis Trust | SRC003 | Michael A Gray |
| Cystic Fibrosis Trust | SRC013 | Michael A Gray |
| Medical Research Council | MC_PC_15030 | Michael A Gray |
| Fondo Nacional de Desarrollo Científico y Tecnológico | 1221257 | Carlos A Flores |

The funders had no role in study design, data collection and interpretation, or the decision to submit the work for publication.

### Author contributions

Vinciane Saint-Criq, Conceptualization, Data curation, Formal analysis, Investigation, Methodology, Writing – original draft, Writing – review and editing; Anita Guequén, Data curation, Formal analysis, Investigation, Methodology, Writing – review and editing; Amber R Philp, Data curation, Investigation; Sandra Villanueva, Ignacio Fernández-Moncada, Data curation, Formal analysis, Investigation, Methodology; Tábata Apablaza, Livia Delpiano, Cristian Carrasco, Investigation; Agustín Mansilla, Methodology; Iván Ruminot, Formal analysis, Investigation, Methodology, Supervision; Michael A Gray, Conceptualization, Formal analysis, Funding acquisition, Investigation, Supervision, Writing – review and editing; Carlos A Flores, Conceptualization, Data curation, Formal analysis, Funding acquisition, Investigation, Methodology, Supervision, Validation, Writing – original draft, Writing – review and editing

### Author ORCIDs

Livia Delpiano ⓘ http://orcid.org/0000-0002-2319-4456
Carlos A Flores ⓘ http://orcid.org/0000-0002-3813-1909

### Ethics

Unless otherwise stated, all procedures were performed after mice were deeply anesthetized via i.p. injection of 120 mg/kg ketamine and 16 mg/kg xylazine followed by exsanguination. All experimental

procedures were approved by the Centro de Estudios Científicos (CECs) Institutional Animal Care and Use Committee (#2015-02) and are in accordance with relevant guidelines and regulations.

### Decision letter and Author response
Decision letter https://doi.org/10.7554/eLife.75871.sa1
Author response https://doi.org/10.7554/eLife.75871.sa2

## Additional files

### Supplementary files
• Supplementary file 1. Primers for hBECs.
• Supplementary file 2. Primers for mouse.
• Transparent reporting form

### Data availability
All data generated or analysed during this study are included in the manuscript and supporting file.

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
