## [Editor Report]

This paper is of interest to scientists and clinicians within the field of muco-obstructive diseases in the airways, such as cystic fibrosis (CF) and chronic obstructive pulmonary disease (COPD). It identifies the sodium-bicarbonate cotransporter SLC4A4 as a key component of the mechanism by which normal airways prevent the formation of sticky mucus and defend their selves against bacterial and viral infections.

---

## [Decision Letter]

**Decision letter after peer review:**

Thank you for submitting your article "Inhibition of the sodium-dependent HCO3 -transporter SLC4A4, produces a cystic fibrosis-like airway disease phenotype." for consideration by *eLife*. Your article has been reviewed by 3 peer reviewers, including László Csanády as Reviewing Editor and Reviewer #1, and the evaluation has been overseen by Richard Aldrich as the Senior Editor. The following individual involved in review of your submission has agreed to reveal their identity: Hugo de Jonge (Reviewer #2).

Essential revisions:

1. SLC4A4, A5 and A8 are the major NCBTs expressed in the primary hAECs (Figure S1A). The arguments used to exclude A5 and A8 as candidate HCO_3_^-^ importers (l. 298-301) are weak because (i) the fact that transepithelial HCO_3_^-^ transport generates a Isc does not prove that the importer is electrogenic (cf. Cl^-^ secretory currents mediated through the electroneutral NKCC1 importer), and (ii) the localization of SLC4A5 in the apical membrane of renal epithelial cells does not rule out a basolateral localization in airway cells (cf. the opposite polarization of AE2 in intestinal vs. bile duct cells and of CA12 in salivary duct vs. airway surface cells). Does the SLC4A4 inhibitor S0859, at 30 μm concentration, also inhibit A5 or A8? Please comment. Furthermore, the authors report that both SLC4A7 and SLC4A10 had very low expression levels, although in Figure S1A it looks as if the expression of SLC4A7 could be quite reasonable. How did the authors determine relative mRNA abundance and, along similar lines, were the amplified bands digested to confirm their identity? Finally, are other candidate HCO_3_^-^ transporters or channels, e.g. Bestrophins, expressed too in primary hAECs?

2. Figure 1A-D: The size of the S0859-inhibitable Isc (representing a HCO_3_^-^ secretory current) in the hAECs is very small (~1 uAmp/cm2) in comparison with anion current measurements in similar cell models reported previously (e.g. refs. 14, 27: ~20 uAmp/cm2). This is remarkable because the conditions used (Cl^-^ free buffer) is optimal for de-inhibition of NBCe1-B activity that is known to act as a Cli sensor through its 2 GXXXP motifs (Shcheynikov et al. 2015 PNAS 112: E329-37). To judge better about the quality of the hAEC monolayers in culture, one would like to know how the inclusion of Cl^-^, or CFTR activation by forskolin in the perfusion medium (affecting the ASL pH; Figure 2) affects Isc measurements in the Ussing chamber, and to what extent S0859 inhibits forskolin-stimulated HCO_3_^-^ currents.

3. Figure 2, G-J: In the hAEC monolayers SLC4A4 seems to be co-expressed with acetylated tubulin in ciliated cells. However it is unclear how many other cell types (CC10-positive Club cells; ionocytes; goblet cells) are retrieved in these human airway cell cultures. In both mouse bronchi and bronchioles (Suppl. Figure 4C) and in differentiated hAECs in the Welsh lab (ref. 15) CFTR (and pendrin) are expressed mainly in CC10+-secretory (Club) cells, not in ciliated cells. This raises the question whether SLC4A4 is also expressed in human Club cells and whether Club cells rather than ciliated cells are the major sources of HCO_3_^-^ secreted into the ASL.

4. Figure 3 A-C: Can the authors explain why the IBMX/forskolin/cAMP-induced anion current, in contrast to the UTP/ca^2+^-induced current, is not reduced in bicarbonate-free buffer, i.e. does not seem to have a HCO_3_^-^ component? In mouse trachea, most if not all Fsk/cAMP-induced anion secretory current is CaCC-mediated, so there is a priori no reason why forskolin-induced currents would not be reduced (at least in part) in a HCO_3_^-^ free buffer.

5. Figure S3D-F: Assuming that the CaCC inhibitor AO1 completely blocks CaCCs and the IBMX/Fsk-induced Isc, how do the authors explain that subsequent addition of S0859 further inhibits a "basal" HCO_3_^-^ secretory current? Does HCO_3_^-^ exit the cell at the apical side through tonically active CFTR, or through an electrogenic HCO_3_^-^/Cl^-^ exchanger (but pendrin is electroneutral)?

6. Figure 3, D-E, Figure S3A and D, and l. 350: The cause of the transient negative change in Isc elicited by S0859 remains to be identified. Could it be that the compound has transient off-target effects on the intracellular ca^2+^ level, or activates basolateral K^+^ channels? It is remarkable that similar deflections are not seen in human AECs (Figure 1C).

7. How close do the hAECs used in this study recapitulate the airways in situ? For example, are CA12 (cf. ref. 11) or ATP12A (cf. ref. 7) highly expressed in these cultures? This could either be addressed using Q-PCR of markers, e.g. CA12 and ATP12A, or by immunostaining of SLC4A4 in CC10+ cells.

8. The systemic Slc4a4 null-mouse model used in this study is known to suffer from many abnormalities, including severe metabolic acidosis due to pRTA (l. 471-480). Whether the reduced level of plasma HCO_3_^-^ contributes to the lack of HCO_3_^-^ secretion and reduced ASL pH in the trachea of the Slc4a4-/- mouse (l. 475) depends on the affinity of the NBCe1 transporter for HCO_3_^-^ ions. Therefore it is of interest to learn whether the transporter is already saturated at 5 mM HCO_3_^-^, or needs higher serosal HCO_3_^-^ levels, e.g. 24 mM. This could be addressed experimentally or at least discussed.

9. L. 496: I agree that an airway-specific Slc4a4 null mouse model created by Cre-Lox technology would be of further help in studying the pathogenesis of muco-obstructive diseases of the lungs. In particular a comparison of Club cell-specific vs. ciliated cell-specific Slc4a4-/- mouse models would be useful to study the functional importance of SLC4A4 in each cell type in vivo.

10. The authors show changes in ASL pH in hAEC cultures. Could the authors include in their Methods a statement about the time point at which the pH was considered to have settled, i.e. at what time were the Figure 2B data points collected following the addition of S0859 or Fsk? Could the authors also comment on the difference in timescale between the effects on intracellular acidification (Figure 1) vs the effect on ASL pH which seems to take ~2hrs (Figure 2)?

11. It looks as if some effects are not fully reversible (Figure S3G)? Could the authors comment on this?

12. In relation to the central thesis that pH is central in the development of the CF mucus phenotype, is there any evidence of changes to mucus rheology in the hAEC cultures accompanying the changes in pH?

13. The authors pursue an experiment to determine whether there is "an active NCBT under resting conditions". But surely Isc measurements are not really made under resting conditions as the ASL is essentially flooded and the ionic composition on the apical side is not likely to match that of a normal ASL. This will affect ion driving forces etc. Could the authors consider this point?

*Reviewer #1 (Recommendations for the authors):*

1. Figure 1A vs. C: Why did the authors change two parameters at the same time, by switching from Cl^-^ free HCO_3_^-^ containing (in A) to Cl^-^ containing HCO_3_^-^ free solution (in C)? Wouldn't a Cl^-^ free HEPES-buffered solution be a better control for the experiment in A? Also, please indicate the timepoint of basolateral drug addition in panels A, C.

2. Figure 2C: In the cells pretreated with S0859 Fsk exposure seems to induce a transient small acidification. When compared to that acidified value, the ensuing pH recovery seems comparable to that observed for cells not pretreated with S0859. Can the authors comment on the likely mechanism of this phenomenon?

3. Figure 2B, D, F: Please define what the symbols represent. Is it pH at a given (which?) timepoint following drug addition? Or mean pH observed over a given (which?) time interval following drug addition?

4. Figure 2B, D, F: The plotted significance values are confusing. If the error bars on the black and blue dot represent S.E.M., than the difference is unlikely to be significant in either panel. Was the significance calculated for the δ-pH values? If so, it would be helpful to also provide mean+/-S.E.M. δ-pH, and indicate the significance value there.

5. Figure 3C: mean+/-S.E.M. bars are hidden behind the individual data points.

6. Figure 3L-M: in panel L please show the data also for the third bar in panel M.

7. Suppl. Figure 3C, F: It is unclear what the bars represent. E.g., in panel C the two bars seem identical (negative), although in panels A-B S0859 induces a positive current change when added after Fsk (i.e., δ-Isc should be positive for S0859). The same applies for the 2nd and 3rd bars in panel F.

8. Suppl. Figure 3G: Several clarifications are needed here:

(i) The legend says "UTP-induced intracellular acidification" but the trace seems to show S0859-induced acidification.

(ii) The legend says "in bicarbonate buffer or HEPES", but in the trace it is not indicated where the buffer change has occurred.

(iii) Please calibrate the ordinate into pH units for a better comparison with panel H.

*Reviewer #2 (Recommendations for the authors):*

1. SLC4A4, A5 and A8 are the major NCBTs expressed in the primary hAECs (Figure S1A). In my view the arguments used to exclude A5 and A8 as candidate HCO_3_^-^ importers (l. 298-301) are weak because (i) the fact that transepithelial HCO_3_^-^ transport generates a Isc does not prove that the importer is electrogenic (cf. Cl^-^ secretory currents mediated through the electroneutral NKCC1 importer), and (ii) the localization of SLC4A5 in the apical membrane of renal epithelial cells does not rule out a basolateral localization in airway cells (cf. the opposite polarization of AE2 in intestinal vs. bile duct cells and of CA12 in salivary duct vs. airway surface cells). Does the SLC4A4 inhibitor S0859, at 30 μm concentration, also inhibit A5 or A8? Are other candidate HCO_3_^-^ transporters or channels , e.g. Bestrophins, expressed too in primary hAECs?

2. Figure 1A-D: The size of the S0859-inhibitable Isc (representing a HCO_3_^-^ secretory current) in the hAECs is very small (~1 uAmp/cm2) in comparison with anion current measurements in similar cell models reported previously (e.g. refs. 14, 27: ~20 uAmp/cm2). This is remarkable because the conditions used (Cl^-^ free buffer) is optimal for de-inhibition of NBCe1-B activity that is known to act as a Cli sensor through its 2 GXXXP motifs (Shcheynikov et al. 2015 PNAS 112: E329-37). To judge better about the quality of the hAEC monolayers in culture, one would like to know how the inclusion of Cl^-^, or CFTR activation by forskolin (affecting the ASL pH; Figure 2) in the perfusion medium affects Isc measurements in the Ussing chamber, and to what extent S0859 inhibits forskolin-stimulated HCO_3_^-^ currents.

3. Figure 2, G-J: In the hAEC monolayers SLC4A4 seems to be co-expressed with acetylated tubulin in ciliated cells. However it is unclear how many other cell types (CC10-positive Club cells; ionocytes; goblet cells) are retrieved in these human airway cell cultures. In both mouse bronchi and bronchioles (Suppl. Figure 4C) and in differentiated hAECs in the Welsh lab (ref. 15) CFTR (and pendrin) are expressed mainly in CC10+-secretory (Club) cells, not in ciliated cells. This raises the question whether SLC4A4 is also expressed in human Club cells and whether Club cells rather than ciliated cells are the major sources of HCO_3_^-^ secreted into the ASL.

4. Figure 3 A-C: Can the authors explain why the IBMX/forskolin/cAMP-induced anion current, in contrast to the UTP/ca^2+^-induced current, is not reduced in bicarbonate-free buffer, i.e. does not seem to have a HCO_3_^-^ component? In mouse trachea, most if not all Fsk/cAMP-induced anion secretory current is CaCC-mediated, so there is a priori no reason why forskolin-induced currents would not be reduced (at least in part) in a HCO_3_^-^ free buffer.

5. Figure S3D-F: Assuming that the CaCC inhibitor AO1 completely blocks CaCCs and the IBMX/Fsk-induced Isc, how do the authors explain that subsequent addition of S0859 further inhibits a "basal" HCO_3_^-^ secretory current? Does HCO_3_^-^ exits the cell at the apical side through tonically active CFTR, or through an electrogenic HCO_3_^-^/Cl^-^ exchanger (but pendrin is electroneutral)?

6. Figure 3, D-E, Figure S3A and D, and l. 350: The cause of the transient negative change in Isc elicited by S0859 remains to be identified. Could it be that the compound has transient off-target effects on the intracellular ca^2+^ level, or activates basolateral K^+^ channels? It is remarkable that similar deflections are not seen in human AECs (Figure 1C).

7. How close do the hAECs used in this study recapitulate the airways in situ? For example, are CA12 (cf. ref. 11) or ATP12A (cf. ref. 7) highly expressed in these cultures?

8. The systemic Slc4a4 null-mouse model used in this study is known to suffer from many abnormalities, including severe metabolic acidosis due to pRTA (l. 471-480). Whether the reduced level of plasma HCO_3_^-^ contributes to the lack of HCO_3_^-^ secretion and reduced ASL pH in the trachea of the Slc4a4-/- mouse (l. 475) depends on the affinity of the NBCe1 transporter for HCO_3_^-^ ions. Therefore it is of interest to learn whether the transporter is already saturated at 5 mM HCO_3_^-^, or needs higher serosal HCO_3_^-^ levels, e.g. 24 mM.

9. L. 496: I agree that an airway-specific Slc4a4 null mouse model created by Cre-Lox technology would be of further help in studying the pathogenesis of muco-obstructive diseases of the lungs. In particular a comparison of Club cell-specific vs. ciliated cell-specific Slc4a4-/- mouse models would be useful to study the functional importance of SLC4A4 in each cell type in vivo.

In my view the paper could be strengthened further by:

1. In general: addressing the points of discussion raised in the public review and incorporating some of the answers in the paper.

2. Providing additional information about the cell composition, Cl^-^ vs. HCO_3_^-^ currents, and possible roles of Slc4A5 and A8 (both highly expressed) in the hAECs.

3. If available, grow primary hAECs under the conditions described in ref. 15 (enriched in CC10+ Club cells) and determine whether NBCe1-B is expressed in the ciliated cells, in the Club cells or in both.

4. Find out whether Slc4A4 becomes already saturated at 5 mM HCO_3_^-^, or needs higher serum HCO_3_^-^ levels for optimal function.

*Reviewer #3 (Recommendations for the authors):*

1. The authors show some nice amplification of SLC4A bands and report that both SLC4A7 and SLC4A10 had very low expression levels, referring the reader to Figure S1A. However, it looks as if the expression of SLC4A7 could be quite reasonable in this figure. How did the authors determine relative mRNA abundance and, along similar lines, were the amplified bands digested to confirm their identity?

2. The authors provide evidence supporting the expression of several members of the SLC4A family of transporters in human airway epithelial cells. Although S0859 is a blocker of SLC4A4, is it selective for SLC4A4 or is it possible other members of the SLC4A family might be blocked by this drug and contributing to some of the observed effects e.g. ASL pH changes? Can the authors comment on this?

3. The authors show changes in ASL pH in hAEC cultures. Could the authors include in their Methods a statement about the time point at which the pH was considered to have settled, i.e. at what time were the Figure 2B data points collected following the addition of S0859 or Fsk? Could the authors also comment on the difference in timescale between the effects on intracellular acidification (Figure 1) vs the effect on ASL pH which seems to take ~2hrs (Figure 2)?

4. It looks as if some effects are not fully reversible (Figure S3G)? Could the authors comment on this?

5. Direct measurements of ASL volume and pH in the knockout mice would clearly strengthen part of the central thesis, although these measurements are admittedly difficult to perform. However, on a related topic, is there any evidence of changes to mucus rheology in the hAEC cultures accompanying the changes in pH?

6. The authors pursue an experiment to determine whether there is "an active NCBT under resting conditions". But surely Isc measurements are not really made under resting conditions as the ASL is essentially flooded and the ionic composition on the apical side is not likely to match that of a normal ASL. This will affect ion driving forces etc. Could the authors consider this point?

---

## [Author Response]

Essential revisions:1. SLC4A4, A5 and A8 are the major NCBTs expressed in the primary hAECs (Figure S1A). The arguments used to exclude A5 and A8 as candidate HCO_3_^-^ importers (l. 298-301) are weak because (i) the fact that transepithelial HCO_3_^-^ transport generates a Isc does not prove that the importer is electrogenic (cf. Cl^-^ secretory currents mediated through the electroneutral NKCC1 importer), and (ii) the localization of SLC4A5 in the apical membrane of renal epithelial cells does not rule out a basolateral localization in airway cells (cf. the opposite polarization of AE2 in intestinal vs. bile duct cells and of CA12 in salivary duct vs. airway surface cells). Does the SLC4A4 inhibitor S0859, at 30 μm concentration, also inhibit A5 or A8? Please comment. Furthermore, the authors report that both SLC4A7 and SLC4A10 had very low expression levels, although in Figure S1A it looks as if the expression of SLC4A7 could be quite reasonable. How did the authors determine relative mRNA abundance and, along similar lines, were the amplified bands digested to confirm their identity? Finally, are other candidate HCO_3_^-^ transporters or channels, e.g. Bestrophins, expressed too in primary hAECs?

We thank the reviewers for their important comment on the potential of other SLC4A family members to be involved in intracellular as ASL pH regulation in airway epithelial cells. After a careful literature review we could not find any decisive evidence whether S0859 specifically inhibits SLC4A4 and not the other SLC4A bicarbonate co transporters (Ch’en et al. Br J Pharmacol. 2008; Schwab et al. J Physiol. 2005). Moreover, although we did not perform qPCR experiments to measure the levels of mRNA of each SLC4A member, RNA-seq data from another study using epithelial cells from the same donors, grown in the same conditions, revealed that SLC4A4 is the most expressed member, closely followed by SLC4A7, SLC4A8 and SLC4A5 (see Figure 1—figure supplement1). No counts were found for SLCA10. Data from this project are reported in the following publication (Saint-Criq et al. Cells. 2020 doi: 10.3390/cells9092137. PMID: 32967385) and the data are available are accessible through Gene Expression Omnibus (GEO) series accession number GSE154905.

We also report the levels of expression of Bestrophins 1, 3 and 4 in primary hAECs. Bestrophins mRNA levels were 1 to 2 log lower than SLC4A4, A7 and A8. Taken together, we believe there is not strong enough evidence that allows us to rule out other members of the SLC4A family, especially SLC4A7 and SLC4A8. Thus, in the manuscript we have added these data as Figure 1—figure supplement1G, edited the main text, and discussed these results further in the Discussion section (lines 228-231).

2. Figure 1A-D: The size of the S0859-inhibitable Isc (representing a HCO_3_^-^ secretory current) in the hAECs is very small (~1 uAmp/cm2) in comparison with anion current measurements in similar cell models reported previously (e.g. refs. 14, 27: ~20 uAmp/cm2). This is remarkable because the conditions used (Cl^-^ free buffer) is optimal for de-inhibition of NBCe1-B activity that is known to act as a Cli sensor through its 2 GXXXP motifs (Shcheynikov et al. 2015 PNAS 112: E329-37). To judge better about the quality of the hAEC monolayers in culture, one would like to know how the inclusion of Cl^-^, or CFTR activation by forskolin in the perfusion medium (affecting the ASL pH; Figure 2) affects Isc measurements in the Ussing chamber, and to what extent S0859 inhibits forskolin-stimulated HCO_3_^-^ currents.

We thank the reviewer for their comment. In Author response image 1 we provide data representing CFTR activation in the presence of HCO_3_^-^ (HCO3 KRB) and in the absence of Cl^-^ (0Cl^-^). As shown in Author response image 1, CFTR-dependent currents were significantly reduced (> 90 %) in the absence of Cl^-^, consistent with previous reports that HCO_3_^-^ has a much lower permeability (conductance) through CFTR compared to Cl^-^ (Gray et al., Am. J. Physiol.1990; Linsdell et al., JCF, 2009). In addition, the size of the FSK-stimulated Isc in the absence of Cl^-^ is actually very similar to that found by Gorrieri (ref 33) using similar conditions (~2 µAmp/cm^2^- see Figure 4C, no IL4), and NOT 20uAmp/cm2 which was the value with both Cl and HCO3.

**Author response image 1. sa2fig1:** Comparison of HCO_3_^-^ KRB and Cl^-^ free solutions on Isc responses of primary hAECs to amiloride, forskolin, potentiator P5 and CFTRInh_172_. Panel on the left shows a representative trace. Panels on the right show either baseline Isc (top left) or changes in Isc (ΔIsc) after specific inhibitors and agonists. Each symbol represents an independent experiment, using cells from two donors. Bars are means ± SD.

We have also included in the revised manuscript, the effect of S0859 on Fsk-induced HCO_3_^-^ secretion in Ussing chamber experiments (Figure 1, panels A and C). In this setting, S0859 did not affect the Fsk-induced change in I_sc_. This result suggests that there may be other mechanisms that the epithelium uses to accumulate bicarbonate intracellularly in order to maintain Fsk-induced bicarbonate secretion. There are a number of possibilities, including, other NBCs that may not be inhibited by S0859 (see point 1 above), as well as the generation of HCO_3_^-^ via intracellular carbonic anhydrase (CA), when the NBC is inhibited. We did test the effect of inhibiting CA (using acetazolamide), on unstimulated and Fsk-stimulated HCO_3_^-^ secretion in Ussing chamber experiments under Cl^-^ free conditions, but did not observe any effect of ACTZ – see Author response image 2. However, we believe that further investigation into this aspect of the work is beyond the scope of this current manuscript, particularly because the 0Cl^-^ condition itself is nonphysiological, and we do not know the full effect of bathing the epithelia in a Cl^-^ free buffer on (i) ion gradients and (ii) epithelial cell homeostasis in general. This is why we studied the effect of NBC inhibition in a more physiologically relevant condition, i.e. under thin film condition using normal physiological solutions (ASL pH measurements) and a murine model. Overall, we believe our results from using both fully differentiated primary human airway epithelial cells and mouse models, strongly support a pivotal role of SLC4A4 in pH airway homeostasis

**Author response image 2. sa2fig2:** Effect of acetazolamide (ACTZ) on unstimulated and Fsk-stimulated HCO_3_^-^ secretion in Ussing chamber experiments in Cl^-^ free buffer. Upper panels show the kinetics of the effect of ACTZ. Bottom panels are the DMSO, ACTZ and Fsk-induced changes.

3. Figure 2, G-J: In the hAEC monolayers SLC4A4 seems to be co-expressed with acetylated tubulin in ciliated cells. However it is unclear how many other cell types (CC10-positive Club cells; ionocytes; goblet cells) are retrieved in these human airway cell cultures. In both mouse bronchi and bronchioles (Suppl. Figure 4C) and in differentiated hAECs in the Welsh lab (ref. 15) CFTR (and pendrin) are expressed mainly in CC10+-secretory (Club) cells, not in ciliated cells. This raises the question whether SLC4A4 is also expressed in human Club cells and whether Club cells rather than ciliated cells are the major sources of HCO_3_^-^ secreted into the ASL.

The samples in Figure 2-G-J correspond to human lung biopsies and not hAEC monolayers. The data from the Welsh paper (15) comes from hAECs incubated with inflammatory cytokines and not naïve cells. It is also important to mention that in the Welsh paper there was no increase in expression of the *SLC4A4* as seen with Pendrin after the cytokine incubation of hAECs (Figure 5B; Welsh paper), suggesting that bicarbonate secretion in non-ciliated cells might depend on a different basolateral bicarbonate carrier. Welsh data is also different to the data from the Hanrahan lab (Ref 14) and Boucher lab (Ref 41) that show CFTR and Pendrin to be expressed in naïve ciliated cells, the same cells where we observe SLC4A4. The last two mentioned papers supports our immunolocalization results, that there is: an important role for ciliated cells in bicarbonate secretion and is consistent with our observations in ASL pH (Figure 2). Finally, we believe that it is very valuable to share this staining of SLC4A4, as is the only staining of SLC4A4 using native tissue known to the date. We added the following at lane 252-255:

“Even though, CFTR and Pendrin expression increases in secretory cells after cytokine stimulation, SLC4A4 remained unaltered, suggesting that other basolateral HCO_3_^-^ transporter support HCO_3_^-^ secretion in human secretory cells during the inflammatory response(15)”.

4. Figure 3 A-C: Can the authors explain why the IBMX/forskolin/cAMP-induced anion current, in contrast to the UTP/ca^2+^-induced current, is not reduced in bicarbonate-free buffer, i.e. does not seem to have a HCO_3_^-^ component? In mouse trachea, most if not all Fsk/cAMP-induced anion secretory current is CaCC-mediated, so there is a priori no reason why forskolin-induced currents would not be reduced (at least in part) in a HCO_3_^-^ free buffer.

Similar to our results, previous work from the laboratory of Marcus Mall (REF 24;Anagnostopolou et al. 2012 JCI 122(10): 3629-3634; Sup Figure 2) showed that the cAMP-induced current is similar when the experiment is performed in HEPES vs HCO_3_^-^ buffer in mouse tracheas. To our knowledge Shah et al.,2016 (Science. 2016 Jan 29; 351(6272): 503–507) is the only paper where bicarbonate secretion in mouse airway cell cultures has been shown. The size of the current recorded after cAMP stimulation is very small (around 4 µA cm-2) and the recordings were performed in bath solution with no Cl^-^ to isolate the bicarbonate current. It might be possible that due to the small magnitude of the bicarbonate current the difference is not noticeable when using buffers containing Cl^-^ as in our case and in the mentioned paper of M. Mall’s group. We have added this reference in the description of Results Lane:144-145

“but no significant effect on the cAMP-induced anion secretion, or the amiloride-sensitive sodium absorption (Figure 3C) was detected as previously shown(24).”

5. Figure S3D-F: Assuming that the CaCC inhibitor AO1 completely blocks CaCCs and the IBMX/Fsk-induced Isc, how do the authors explain that subsequent addition of S0859 further inhibits a "basal" HCO_3_^-^ secretory current? Does HCO_3_^-^ exit the cell at the apical side through tonically active CFTR, or through an electrogenic HCO_3_^-^/Cl^-^ exchanger (but pendrin is electroneutral)?

Previous experiments performed in tracheas of mice showed that the genetic silencing of Tmem16a doesn’t affect the cAMP-induced anion seretion but reduces by nearly 50% the Ca^+2^induced anionic secretion (Rock et al., J Biol Chem. 2009 May 29; 284(22): 14875–14880; Ousingsawat et al., J Biol Chem. 2009 Oct 16;284(42):28698-703). Our own exploration of anion secretion in mouse airways and hAECs demonstrated that CaCCinhA01 inhibits both TMEM16A and CFTR (Ref 68), suggesting that CaCCinhA01 affects cAMP-induced secretion. Our interpretation of those and our present results is that in mouse trachea there is an unidentified anion channel that participates in anion secretion, but such statement (and aim) is beyond the scope of the present work and would require additional experiments, including the breeding a double null Cftr/Tmem16a mouse, to test the hypothesis.

6. Figure 3, D-E, Figure S3A and D, and l. 350: The cause of the transient negative change in Isc elicited by S0859 remains to be identified. Could it be that the compound has transient off-target effects on the intracellular ca^2+^ level, or activates basolateral K^+^ channels? It is remarkable that similar deflections are not seen in human AECs (Figure 1C).

Other authors using hAEC have also observed this transient change in Isc (REF 25; Figure 5 A,B) that we cited in the text lane 163. Is possible to observe such negative deflection in the hAECs monolayer used by us in Figure 1A. As the reviewer suggests S0859 has some off targets so, we have added this information and possibility at Lane 161-164:

“Of note, the addition of S0859 to the tracheas induced a fast and transient negative change in Isc as observed in Figure 3D and E and Figure 3-figure 3 supplement 2A and D, that has been also observed in human cells (25), and that might be due to off-targets of the blocker like other SLC4A or SLC16A transporters as previously described (26-28).”

7. How close do the hAECs used in this study recapitulate the airways in situ? For example, are CA12 (cf. ref. 11) or ATP12A (cf. ref. 7) highly expressed in these cultures? This could either be addressed using Q-PCR of markers, e.g. CA12 and ATP12A, or by immunostaining of SLC4A4 in CC10+ cells.

This is a very important question. To fully answer this we would have to compare levels of expression of multiple airway epithelial cell markers between our cultures and native human lung tissue. At this stage, we are not able to obtain freshly isolated cells from non-CF donors to undertake this comparison. However, as stated previously, we have RNAseq data from a different study, and Author response image 3 shows the normalized counts for some airway epithelial ion channels and transporters. As shown in the figure, and published before, ATP12A is highly expressed in our primary cells (REF 22), whereas expression of CA12 is ~ 2-log units lower, and similar to CFTR and β ENaC. Recent work from Kim et al.,2021 (REF 11), compared relative CA12 mRNA expression between freshly isolated cells from CF bronchus to cultured CF and non-CF ALI grown bronchial cells. They showed that CA12 levels were ~ 2-fold higher in freshly isolated CF cells compared to CF cultured cells, but similar to non-CF cultured cells. However, the expression levels were very donor-dependent in cultured cells, but this issue was not investigated in freshly isolated cells. Overall, this study showed that cultured ALI grown cells were a good model of the airways in situ*.* However, it is indeed very important to optimize growth conditions as these can affect mRNA and protein levels as well as the function of channels, transporters and enzymes involved in ASL pH homeostasis, as we have previously described (REF 17).

**Author response image 3. sa2fig3:** Normalised counts of ion channels and transporters expressed in primary human airway epithelial cells. Each symbol represents one donor. Bars are means ± SD.

8. The systemic Slc4a4 null-mouse model used in this study is known to suffer from many abnormalities, including severe metabolic acidosis due to pRTA (l. 471-480). Whether the reduced level of plasma HCO_3_^-^ contributes to the lack of HCO_3_^-^ secretion and reduced ASL pH in the trachea of the Slc4a4-/- mouse (l. 475) depends on the affinity of the NBCe1 transporter for HCO_3_^-^ ions. Therefore it is of interest to learn whether the transporter is already saturated at 5 mM HCO_3_^-^, or needs higher serosal HCO_3_^-^ levels, e.g. 24 mM. This could be addressed experimentally or at least discussed.

We thank the reviewer for this comment. Effectively, our own experiments (REF 56) demonstrate that NBCe1 is a low affinity bicarbonate transporter that is saturated at HCO_3_^-^ 3 mM, a concentration significantly lower than that observed in the systemic Slc4a4 null-mouse (close to 5 mM; REF 54). Thus the reduction in plasma bicarbonate as observed in pRTA (16-20 mM), or in the KO mouse, largely exceeds the concentration at which the SLC4A4 transporter is saturated (Palmer et al., (2021) Adv Ther 38(2):949-968). To include this observation we have rephrased the Discussion including references at Lane 297-299 “Even though, decreased plasma HCO_3_^-^ is observed in the Slc4a4-/- mice (5.3 ± 0.5 mM) (54) is unlikely that reduced HCO_3_^-^ availability influenced mucus accumulation in the animals, as we have previously determined that the transporter is fully saturated at 3 mM HCO_3_^-^ (56).”

9. L. 496: I agree that an airway-specific Slc4a4 null mouse model created by Cre-Lox technology would be of further help in studying the pathogenesis of muco-obstructive diseases of the lungs. In particular a comparison of Club cell-specific vs. ciliated cell-specific Slc4a4-/- mouse models would be useful to study the functional importance of SLC4A4 in each cell type in vivo.

We are currently breeding the Club cell specific null mouse but the results will only be available in a year or so.

10. The authors show changes in ASL pH in hAEC cultures. Could the authors include in their Methods a statement about the time point at which the pH was considered to have settled, i.e. at what time were the Figure 2B data points collected following the addition of S0859 or Fsk? Could the authors also comment on the difference in timescale between the effects on intracellular acidification (Figure 1) vs the effect on ASL pH which seems to take ~2hrs (Figure 2)?

We thank the reviewer for their comments. We have now added the following statement to the Methods section: Lane: 386-391;

“Changes in ASL pH (ΔASLpH) were calculated by averaging five time points (average pH over 25 min) before and 2hrs after the addition of the chemicals (Fsk/S0859).”

However, we would like to point out that the time taken to reach steady-state was actually less than 2hrs after the addition of Fsk. For comparison purposes, a better representation of the time taken to reach steady-state is to calculate the t_1/2_ and we have presented these values in Figure 2. The t_1/2_ for the effect of S0859 on resting and Fsk-stimulated ASL pH was 46 mins and 14 mins, respectively, whereas the t_1/2_ for the effect of Fsk on resting and S0859inhibited ASL pH was 25 mins and 31 mins, respectively.

With regard to the different time scales between the pH_i_ and ASL pH experiments, we would like to point out that making a ‘direct’ comparison is complicated for a number of reasons. (1) The experimental technique and conditions used to assess pH changes differed between the two measurements. For ASL pH, the cell cultures were maintained at the air-liquid interface with differentiation medium only on the basolateral side of the cultures, which was not perfused. Small aliquots of stock solutions of chemicals were manually added to the basolateral solution. For the pH_i_ experiments, the cell cultures were submerged in buffer and continuously perfused apically and basolaterally. Exchange times for solution changes in these pHi expts were ~ 1 min. (2) For the pH_i_ experiments, intracellular acidification was induced by increasing CO_2_, whereas for the ASL pH expts, changes in ASL pH were induced by adding agonists/inhibitors to the nonperfused basolateral compartment. In separate (new) experiments, when CO_2_ levels were acutely increased from 0.1 % to 5.0 %, ASL acidified with a t_1/2_ of 6.7 mins (n=3 – see Author response image 5), considerably faster than the agonist-induced changes. (3) For the ASL pH experiments, although the apical surface was washed 24hr before experiments started, these epithelial cells produce mucus, which accumulates in the ASL over the 24 hr period, and very likely increases the buffering capacity of the luminal surface, thereby slowing pH changes. Indeed, Kim et al., (JEM, 2021) recently showed that adding purified mucin to washed ALI cultures significantly slowed ASL pH changes induced by CO_2_ removal by ~ 80%. (4) The cultured cells also express multiple carbonic anhydrases (Author response image 5 – Results from the RNA-seq study), and other transporters that will also impact the kinetics of responses. We are not aware of any reports which have directly compared the activity of cytosolic carbonic anhydrases versus extracellular CAs, but this is also an important factor to consider, as CA activity has pronounced effects on pH kinetics (Kim et al., JEM. 2021; REF 11). Overall, both technically, and experimentally ASL pH changes would be expected to take longer to elicit a response.

**Author response image 4. sa2fig4:** A. Effect of increasing CO_2_ concentration on ASL pH (black line) of hAECs (n=3, mean ± SD). Exponential decay nonlinear regression analysis (red dashed line) and simple linear analysis (blue line) were performed on the ASL pH values from the change in CO_2_ concentration. B. Average rate of CO_2_induced acidification of 3 independent experiments measured as the slope of the linear regression analysis.

**Author response image 5. sa2fig5:** Normalised counts of carbonic anhydrase (CA) family members expressed in primary human airway epithelial cells. Each symbol represents one donor. Bars are means ± SD.

11. It looks as if some effects are not fully reversible (Figure S3G)? Could the authors comment on this?

This might be because the binding of the inhibitor to the SLC4A4 transporter cannot be fully washed off. However we found reversibility to be variable between experiments. We have changed the figure for a different trace where it is possible to observe that the effect of S0859 is more reversible than in the original trace. Figure 3—figure supplement 2G.

12. In relation to the central thesis that pH is central in the development of the CF mucus phenotype, is there any evidence of changes to mucus rheology in the hAEC cultures accompanying the changes in pH?

We have not measured mucus rheology in hAECs after treatment with S0859. However, Tang et al. 2016 (JCI 126(3); 879-891; REF 49). They observed increased viscosity of the ASL (including mucins) obtained from cell cultures of human AECs and pig AECs affected by CF (FIGURE 2). The increased viscosity parallels with a lower pH compared with cultured cells isolated from non-CF pigs (FIGURE 4). We have added this evidence and corresponding reference in the Discussion at Lane 283-284

“For example, hAEC from CF patients produce acidic ASL and mucus that is more viscous than in cells from non-CF donors (49).”

13. The authors pursue an experiment to determine whether there is "an active NCBT under resting conditions". But surely Isc measurements are not really made under resting conditions as the ASL is essentially flooded and the ionic composition on the apical side is not likely to match that of a normal ASL. This will affect ion driving forces etc. Could the authors consider this point?

We agree with the reviewer that these conditions did not represent resting conditions. We have amended the text and replaced ‘resting’ by ‘unstimulated’

Reviewer #1 (Recommendations for the authors):1. Figure 1A vs. C: Why did the authors change two parameters at the same time, by switching from Cl^-^ free HCO_3_^-^ containing (in A) to Cl^-^ containing HCO_3_^-^ free solution (in C)? Wouldn't a Cl^-^ free HEPES-buffered solution be a better control for the experiment in A? Also, please indicate the timepoint of basolateral drug addition in panels A, C.

These conditions were selected in order to specifically isolate Cl^-^ and HCO_3_^-^ currents. Although a Cl^-^ free HEPES buffered solution would have indeed been a complementary control condition, we wanted to specifically evaluate the impact of S0859 on the HCO_3_^-^ current alone and Cl^-^ current, independently of the other anion. Dotted lines have been added to Figure 1A and 1C to indicate the drug addition.

2. Figure 2C: In the cells pretreated with S0859 Fsk exposure seems to induce a transient small acidification. When compared to that acidified value, the ensuing pH recovery seems comparable to that observed for cells not pretreated with S0859. Can the authors comment on the likely mechanism of this phenomenon?

We believe the transient acidification seen in the S0859 treated cell was an artefact, as this did not occur in all experiments, as seen in Author response image 6. This figure shows individual ASL pH recordings in S0859-treated hAECs obtained from 9 independent experiments. We can see that only two experiments (panels D and E) showed a transient acidification following forskolin addition to the basolateral medium. Note that the peaks observed at the time of addition of forskolin are due to the drop in CO_2_ (from 5% to air) when the plates were removed from the plate-reader in order to treat the cells.

If it isn’t an artefact, this acidification could be due to concurrent Fsk-stimulated H^+^ secretion (as shown by Shah et al., (2016) in pig and human airway epithelial cells in the absence of HCO3/CO_2_). In DMSO treated cells, this would be masked by HCO_3_^-^ secretion (driven by accumulated intracellular HCO_3_^-^). In S0859-treated cells, cytoplasmic HCO_3_^-^ accumulation is decreased, and thus H^+^ secretion into the ASL would be transiently visible.

**Author response image 6. sa2fig6:** Individual ASL pH measurements in S0859-treated primary hAECs show that forskolin (FSK) only induced a transient acidification, before the subsequent alkalinisation, in 2 out of 9 independent experiments (panels D and E). Note that the peaks observed at the time of addition of the forskolin are due to the drop in CO_2_ when the plates were removed from the plate-reader in order to treat the cells.

3. Figure 2B, D, F: Please define what the symbols represent. Is it pH at a given (which?) timepoint following drug addition? Or mean pH observed over a given (which?) time interval following drug addition?

We have added the following sentence to the figure legend as well as in the methods section. Lane 386-387:

“Changes in ASL pH (ΔASLpH) were calculated by averaging five time points (average pH over 25 min) before and 2hrs after the addition of the molecules (Fsk/S0859)”

4. Figure 2B, D, F: The plotted significance values are confusing. If the error bars on the black and blue dot represent S.E.M., than the difference is unlikely to be significant in either panel. Was the significance calculated for the δ-pH values? If so, it would be helpful to also provide mean+/-S.E.M. δ-pH, and indicate the significance value there.

The statistical analysis was performed on ASL pH (panels B and F) and ΔASLpH (panel D) values. Paired t-tests were chosen as the changes in ASL pH were dependent on the donor cells and passage number. We provide Author response table 1, the raw data and statistical analysis obtained using GraphPad Prism.

**Author response table 1. sa2table1:** 

Figure 2 panel B		Baseline	S0859		Figure 2 panel D		FSK	S0859 + FSK		Figure 2 panel F		FSK	FSK + S0859	
20170602 Donor1 P2	6,52	6,52	6,46	6,46	20170602 Donor1 P2	0,066	0,066	0,034	0,034	20170602 Donor1 P2	6,72	6,72	6,68	6,68
20170607 Donor 1 P4	6,84	6,84	6,78	6,78	20170607 Donor 1 P4	0,092	0,092	0,100	0,100	20170607 Donor 1 P4	6,86	6,86	6,84	6,84
20170620 Donor 2 P2	7,13	7,13	7,01	7,01	20170620 Donor 2 P2	0,055	0,055	-0,029	-0,029	20170620 Donor 2 P2	7,28	7,28	7,25	7,25
20170621 Donor 3 P2	6,94	6,94	6,86	6,86	20170621 Donor 3 P2	0,090	0,090	-0,001	-0,001	20170621 Donor 3 P2	7,12	7,12	7,10	7,10
20170627 Donor 2 P3	7,22	7,22	7,22	7,22	20170627 Donor 2 P3	0,125	0,125	0,033	0,033	20170627 Donor 2 P3	7,35	7,35	7,24	7,24
20170628 Donor 3P3	6,62	6,62	6,56	6,56	20170628 Donor 3P3	0,125	0,125	0,044	0,044	20170628 Donor 3P3	6,77	6,77	6,70	6,70
20190605	7,02	7,02	6,93	6,93	20190605	0,141	0,141	0,074	0,074	20190605	7,28	7,28	7,21	7,21
20190618	6,87	6,87	6,67	6,67	20190618	0,077	0,077	-0,030	-0,030	20190618	7,09	7,09	7,03	7,03
20190620	6,43	6,43	6,38	6,38	20190620	0,079	0,079	-0,013	-0,013	20190620	6,52	6,52	6,47	6,47
														
Table Analyzed	means S0859 ASL pH baseline				Table Analyzed	means S0859 ASL pH Fsk				Table Analyzed	means ASL pH Fsk S0859			
														
Column C	S0859				Column C	S0859 + FSK				Column C	FSK + S0859			
vs.	vs,				vs.	vs,				vs.	vs,			
Column B	Baseline				Column B	FSK				Column B	FSK			
														
Paired t test					Paired t test					Paired t test				
P value	0,003				P value	<0,001				P value	<0,001			
P value summary	**				P value summary	***				P value summary	***			
Significantly different (P < 0.05)?	Yes				Significantly different (P < 0.05)?	Yes				Significantly different (P < 0.05)?	Yes			
One- or two-tailed P value?	Two-tailed				One- or two-tailed P value?	Two-tailed				One- or two-tailed P value?	Two-tailed			
t, df	t=4,273, df=8				t, df	t=5,832, df=8				t, df	t=5,217, df=8			
Number of pairs	9				Number of pairs	9				Number of pairs	9			
														
How big is the difference?					How big is the difference?					How big is the difference?				
Mean of differences (C – B)	-0,07895				Mean of differences (C – B)	-0,07089				Mean of differences (C – B)	-0,05307			
SD of differences	0,05543				SD of differences	0,03647				SD of differences	0,03052			
SEM of differences	0,01848				SEM of differences	0,01216				SEM of differences	0,01017			
95% confidence interval	-0,1216 to -0,03634				95% confidence interval	-0,09893 to -0,04286				95% confidence interval	-0,07653 to -0,02961			
R squared (partial eta squared)	0,6954				R squared (partial eta squared)	0,8096				R squared (partial eta squared)	0,7728			
														
How effective was the pairing?					How effective was the pairing?					How effective was the pairing?				
Correlation coefficient (r)	0,9794				Correlation coefficient (r)	0,6002				Correlation coefficient (r)	0,9949			
P value (one tailed)	<0,001				P value (one tailed)	0,044				P value (one tailed)	<0,001			
P value summary	****				P value summary	*				P value summary	****			
Was the pairing significantly effective?	Yes				Was the pairing significantly effective?	Yes				Was the pairing significantly effective?	Yes			

5. Figure 3C: mean+/-S.E.M. bars are hidden behing the individual data points.

Individual points were reduced in size and S.E.M. lines width increased.

6. Figure 3L-M: in panel L please show the data also for the third bar in panel M.

Done.

7. Suppl. Figure 3C, F: It is unclear what the bars represent. E.g., in panel C the two bars seem identical (negative), although in panels A-B S0859 induces a positive current change when added after Fsk (i.e., δ-Isc should be positive for S0859). The same applies for the 2nd and 3rd bars in panel F.

The values in Figure 3—figure supplement 2C represent the negative current elicited after IBMX/FSK and then the amount of such negative current inhibited by S0859. Is the same logic as is plotted in Figure 3C for the amiloride-sensitive current that reflects the inhibition of Na^+^absorption, or in both cases, the amount of the negative current that is inhibited by the blockers.

8. Suppl. Figure 3G: Several clarifications are needed here:(i) The legend says "UTP-induced intracellular acidification" but the trace seems to show S0859-induced acidification.

The whole legend is “UTP-induced intracellular acidification in the presence of 30 µM S0859”, and we aim to show the effect of SLC4A4 blocking in UTP-induced intracellular acidification.

(ii) The legend says "in bicarbonate buffer or HEPES", but in the trace it is not indicated where the buffer change has occurred.

The reference to HEPES was a mistake of due to copy and paste from other figures and was taken out from the legend.

(iii) Please calibrate the ordinate into pH units for a better comparison with panel H.

Done.

Reviewer #2 (Recommendations for the authors):1. SLC4A4, A5 and A8 are the major NCBTs expressed in the primary hAECs (Figure S1A). In my view the arguments used to exclude A5 and A8 as candidate HCO_3_^-^ importers (l. 298-301) are weak because (i) the fact that transepithelial HCO_3_^-^ transport generates a Isc does not prove that the importer is electrogenic (cf. Cl^-^ secretory currents mediated through the electroneutral NKCC1 importer), and (ii) the localization of SLC4A5 in the apical membrane of renal epithelial cells does not rule out a basolateral localization in airway cells (cf. the opposite polarization of AE2 in intestinal vs. bile duct cells and of CA12 in salivary duct vs. airway surface cells). Does the SLC4A4 inhibitor S0859, at 30 μm concentration, also inhibit A5 or A8? Are other candidate HCO_3_^-^ transporters or channels , e.g. Bestrophins, expressed too in primary hAECs?

Answered in the essential revisions section

2. Figure 1A-D: The size of the S0859-inhibitable Isc (representing a HCO_3_^-^ secretory current) in the hAECs is very small (~1 uAmp/cm2) in comparison with anion current measurements in similar cell models reported previously (e.g. refs. 14, 27: ~20 uAmp/cm2). This is remarkable because the conditions used (Cl^-^ free buffer) is optimal for de-inhibition of NBCe1-B activity that is known to act as a Cli sensor through its 2 GXXXP motifs (Shcheynikov et al. 2015 PNAS 112: E329-37). To judge better about the quality of the hAEC monolayers in culture, one would like to know how the inclusion of Cl^-^, or CFTR activation by forskolin (affecting the ASL pH; Figure 2) in the perfusion medium affects Isc measurements in the Ussing chamber, and to what extent S0859 inhibits forskolin-stimulated HCO_3_^-^ currents.

Answered in the essential revisions section

3. Figure 2, G-J: In the hAEC monolayers SLC4A4 seems to be co-expressed with acetylated tubulin in ciliated cells. However it is unclear how many other cell types (CC10-positive Club cells; ionocytes; goblet cells) are retrieved in these human airway cell cultures. In both mouse bronchi and bronchioles (Suppl. Figure 4C) and in differentiated hAECs in the Welsh lab (ref. 15) CFTR (and pendrin) are expressed mainly in CC10+-secretory (Club) cells, not in ciliated cells. This raises the question whether SLC4A4 is also expressed in human Club cells and whether Club cells rather than ciliated cells are the major sources of HCO_3_^-^ secreted into the ASL.

Answered in the essential revisions section.

4. Figure 3 A-C: Can the authors explain why the IBMX/forskolin/cAMP-induced anion current, in contrast to the UTP/ca^2+^-induced current, is not reduced in bicarbonate-free buffer, i.e. does not seem to have a HCO_3_^-^ component? In mouse trachea, most if not all Fsk/cAMP-induced anion secretory current is CaCC-mediated, so there is a priori no reason why forskolin-induced currents would not be reduced (at least in part) in a HCO_3_^-^ free buffer.

Answered in the essential revisions section.

5. Figure S3D-F: Assuming that the CaCC inhibitor AO1 completely blocks CaCCs and the IBMX/Fsk-induced Isc, how do the authors explain that subsequent addition of S0859 further inhibits a "basal" HCO_3_^-^ secretory current? Does HCO_3_^-^ exits the cell at the apical side through tonically active CFTR, or through an electrogenic HCO_3_^-^/Cl^-^ exchanger (but pendrin is electroneutral)?

Answered in the essential revisions section

6. Figure 3, D-E, Figure S3A and D, and l. 350: The cause of the transient negative change in Isc elicited by S0859 remains to be identified. Could it be that the compound has transient off-target effects on the intracellular ca^2+^ level, or activates basolateral K^+^ channels? It is remarkable that similar deflections are not seen in human AECs (Figure 1C).

Answered in the essential revisions section

7. How close do the hAECs used in this study recapitulate the airways in situ? For example, are CA12 (cf. ref. 11) or ATP12A (cf. ref. 7) highly expressed in these cultures?8. The systemic Slc4a4 null-mouse model used in this study is known to suffer from many abnormalities, including severe metabolic acidosis due to pRTA (l. 471-480). Whether the reduced level of plasma HCO_3_^-^ contributes to the lack of HCO_3_^-^ secretion and reduced ASL pH in the trachea of the Slc4a4-/- mouse (l. 475) depends on the affinity of the NBCe1 transporter for HCO_3_^-^ ions. Therefore it is of interest to learn whether the transporter is already saturated at 5 mM HCO_3_^-^, or needs higher serosal HCO_3_^-^ levels, e.g. 24 mM.

Answered in the essential revisions section.

9. L. 496: I agree that an airway-specific Slc4a4 null mouse model created by Cre-Lox technology would be of further help in studying the pathogenesis of muco-obstructive diseases of the lungs. In particular a comparison of Club cell-specific vs. ciliated cell-specific Slc4a4-/- mouse models would be useful to study the functional importance of SLC4A4 in each cell type in vivo.

Answered in the essential revisions section.

Reviewer #3 (Recommendations for the authors):1. The authors show some nice amplification of SLC4A bands and report that both SLC4A7 and SLC4A10 had very low expression levels, referring the reader to Figure S1A. However, it looks as if the expression of SLC4A7 could be quite reasonable in this figure. How did the authors determine relative mRNA abundance and, along similar lines, were the amplified bands digested to confirm their identity?

We have answered this question in the Essential Revisions section, and have amended the text and further discussed these results.

2. The authors provide evidence supporting the expression of several members of the SLC4A family of transporters in human airway epithelial cells. Although S0859 is a blocker of SLC4A4, is it selective for SLC4A4 or is it possible other members of the SLC4A family might be blocked by this drug and contributing to some of the observed effects e.g. ASL pH changes? Can the authors comment on this?

This is now discussed in lines 228-231.

3. The authors show changes in ASL pH in hAEC cultures. Could the authors include in their Methods a statement about the time point at which the pH was considered to have settled, i.e. at what time were the Figure 2B data points collected following the addition of S0859 or Fsk? Could the authors also comment on the difference in timescale between the effects on intracellular acidification (Figure 1) vs the effect on ASL pH which seems to take ~2hrs (Figure 2)?

Details about the ASL pH measurements and analysis have been added to the Methods section (lines 386-391) and in the legend of Figure 2

4. It looks as if some effects are not fully reversible (Figure S3G)? Could the authors comment on this?

Answered in the Essential Revisions section.

5. Direct measurements of ASL volume and pH in the knockout mice would clearly strengthen part of the central thesis, although these measurements are admittedly difficult to perform. However, on a related topic, is there any evidence of changes to mucus rheology in the hAEC cultures accompanying the changes in pH?

Answered in the Essential Revisions section.